# GUIDED QUERY REFINEMENT: MULTIMODAL HYBRID RETRIEVAL WITH TEST-TIME OPTIMIZATION

**Omri Uzan**[1]* **Asaf Yehudai**[2,3] **Roi Pony**[2] **Eyal Shnarch**[2] **Ariel Gera**[2]
[1]Stanford University  [2]IBM Research  [3]The Hebrew University of Jerusalem

uzan@stanford.edu

## ABSTRACT

Multimodal encoders have pushed the boundaries of visual document retrieval, matching textual query tokens directly to image patches and achieving state-of-the-art performance on public benchmarks. Recent models relying on this paradigm have massively scaled the sizes of their query and document representations, presenting obstacles to deployment and scalability in real-world pipelines. Furthermore, purely vision-centric approaches may be constrained by the inherent modality gap still exhibited by modern vision-language models. In this work, we connect these challenges to the paradigm of hybrid retrieval, investigating whether a lightweight dense text retriever can enhance a stronger vision-centric model. Existing hybrid methods, which rely on coarse-grained fusion of ranks or scores, fail to exploit the rich interactions within each model's representation space. To address this, we introduce Guided Query Refinement (GQR), a novel test-time optimization method that refines a primary retriever's query embedding using guidance from a complementary retriever's scores. Through extensive experiments on visual document retrieval benchmarks, we demonstrate that GQR allows vision-centric models to match the performance of models with significantly larger representations, while being up to 14x faster and requiring 54x less memory. Our findings show that GQR effectively pushes the Pareto frontier for performance and efficiency in multimodal retrieval. We release our code here.

## 1 INTRODUCTION

Visual document retrieval is the task of returning relevant documents – typically PDFs containing figures, tables, and other visual elements – in response to a textual query (Mathew et al., 2021b;a; Li et al., 2024; Zhu et al., 2022; Faysse et al., 2025). To tackle this task, neural retrieval pipelines often follow a text-centric approach, relying on OCR or vision-language models to convert source documents into textual chunks, and then constructing an index using semantic text encoders (Karpukhin et al., 2020; Tanaka et al., 2021). An alternative, vision-centric, approach relies instead on multimodal encoder models. Building on the ColBERT (Khattab & Zaharia, 2020) late-interaction approach, *ColPali*-based (Faysse et al., 2025) encoders operate directly on image patches, and yield multi-vector embedding representations of images and queries.

While this approach achieves state-of-the-art results on public benchmarks of visual document retrieval[1] (Macé et al., 2025), open challenges within this paradigm remain. First, to pursue state-of-the-art performance, recent late-interaction multimodal retrievers[2] massively scale the length and dimensionality of query and document representations. This can incur a substantial latency and storage overhead, hindering the ability to provide an efficient and scalable solution. For example, LLAMA-NEMORETRIEVER-COLEMBED-3B represents each document page with 10 MB of memory (Xu et al., 2025), three orders of magnitude more than single-vector dense retrievers (Table 4). Secondly, a vision-centric approach for matching textual queries to textually rich documents may

---

*This work was conducted at IBM Research.

[1]https://huggingface.co/blog/manu/vidore-v2

[2]Henceforth, we use "retrievers" and "encoders" interchangeably. For additional background on these emerging retrieval paradigms, refer to Appendix A.

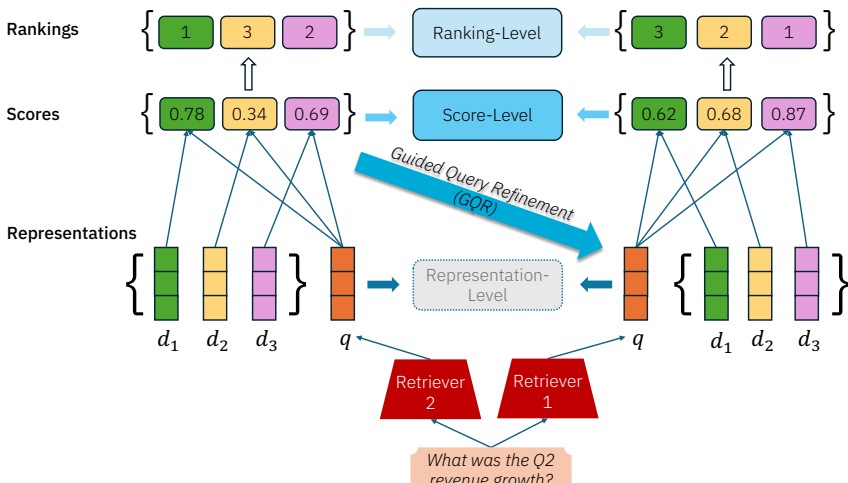

Figure 1: **Hybrid retrieval methods**. Aggregating the outputs of two retrievers is typically done at the level of ranks (§2.1) or scores (§2.2). Utilizing the information of both representations effectively and efficiently is difficult to achieve in practice. Here we propose a novel approach of *Guided Query Refinement (GQR)*, using similarity scores from an complementary retriever (*left*) at test time, to inform the query representation of a primary retriever (*right*).

be limited by the substantial modality gap (Clavié & Brand, 2025; Li et al., 2025; Role et al., 2025) exhibited by modern vision-language models. These gaps motivate exploring complementary approaches for improving the performance of multimodal encoders.

An early concept in the application of neural retrievers has been that of *hybrid retrieval* (dos Santos et al., 2015; Kuzi et al., 2020), where the outputs of different retrievers are aggregated at the level of ranks or query-document similarity scores (see Figure 1) to obtain the final list of retrieved documents. Hybrid retrieval (or *hybrid search*) most commonly refers to the combination of a neural semantic text retriever with a sparse lexical representation (e.g., BM25). More broadly, it reflects the notion that models relying on different types of representations can capture complementary aspects of the data, which can in turn be leveraged to boost the overall performance on the task.

In this work, we seek to connect these two threads, testing whether the paradigm of hybrid retrieval can complement modern multimodal encoders. Dense text retrievers typically have low online latencies and incur a small storage footprint, and they provide a uni-modal signal between the query and the document representation. Thus, hybrid retrieval between text and image encoders emerges as a natural candidate for enhancing the performance of a multimodal retrieval system. However, in this work we argue that standard hybrid retrieval methods rely on a rather coarse-grained view of the perspective of each retriever – they cannot utilize the rich query-document interactions within the model representation space (Figure 1).

Aiming to harness this untapped potential of hybrid retrieval, in this work we propose *Guided Query Refinement (GQR)* (Figure 2), a novel approach for aggregating retriever outputs. Given a query at test time, GQR iteratively optimizes the query representations of a primary retriever with gradient descent, using similarity scores from a complementary retriever. The refined query representation softly incorporates the complementary retriever's signal, remaining subject to the query–document interactions in the primary retriever space. This updated query embedding is then used to score the documents and return an updated document list. Notably, GQR is architecture-agnostic and can be used to optimize both single- and multi-vector query embeddings.

We conduct extensive experiments on established visual retrieval benchmarks, evaluating nine pairs of state-of-the-art vision and text retrievers and comparing GQR to standard hybrid retrieval approaches. Our results (§3.2) demonstrate consistent gains for models using GQR over base models and other hybrid baselines. Despite the fact that text-centric models achieve lower performance on the task, we find that the complementary signal they provide through GQR proves useful for *ColPali*-based models. On ViDoRe 2, COLNOMIC-EMBED-MULTIMODAL-7B with GQR is nearly

on par with LLAMA-NEMORETRIEVER-COLEMBED-3B, while being $\approx \times 14$ faster and requiring $\approx \times 54$ less memory, and outperforms it while being $\approx \times 7$ faster and requiring $\approx \times 24$ less memory. Our results and analysis establish that *ColPali*-based methods using GQR are on the latency and memory Pareto-fronts on the task of visual document retrieval.

## 2 METHODS

Hybrid retrieval variants can be organized into three conceptual levels, reflecting the granularity in which test-time aggregation is performed (Figure 1): the level of *document rankings*, the level of query-document *similarity scores*, or the level of *embedding representations*. The earlier the aggregation, the more information is available, and the more informative the exchange between models can be; however, richer information also increases the burden of normalization and geometrical alignment across spaces.

We begin this section by outlining prominent methods at each level (§2.1, §2.2). We explain that while early representation-level aggregation could be desired due to its richness, it is difficult to achieve in practice. We then present our method, *Guided Query Refinement (GQR)*, which lies between the levels of scores and of representations (§2.3) and provide the motivation for it (§2.4).

**Notations.** Given a query $q$ and retriever $m$, we denote the representation of $q$ by $m$ as $e_m^q$. Similarly, given a set of documents $D = \{d_i\}_{i=1}^N$, we have $e_m^{d_i}$ for all $d_i \in D$. Document relevance to the query is estimated using a similarity score $s_m(q, d_i)$ between the representations $e_m^q$ and $e_m^{d_i}$, typically via cosine similarity (Equation 3) or MaxSim (Equation 4).

$\pi_m(q)$ denotes the list of the documents returned by retriever $m$ for query $q$. $K$ is the length of the retrieved list of documents, and we assume it is constant across retrievers. $\text{rank}_m(d)$ is the 1-indexed position of $\pi_m(q)$ after sorting $\pi_m(q)$ by the scores $s_m(q, \cdot)$ in descending order. If $d \notin \pi_m(q)$, then $\text{rank}_m(q, d) = K+1$.

Finally, while the formulation is general and applies to any number of retrievers $M$, in this work we focus on the case of $M = 2$.

### 2.1 RANKING-LEVEL AGGREGATION

Ranking-level aggregation is the simplest form of information exchange between retrievers: each query and document pair is reduced to a single integer rank. While limited in its expressivity, it requires no extra normalizations or alignments, and is therefore widely used in production pipelines[3].

**Reciprocal Rank Fusion (RRF).** RRF (Cormack et al., 2009) combines ranked lists by weighting each item based on the reciprocal of its rank. The RRF constant $\kappa > 0$ dampens the impact of very high ranks and controls how much credit is given to mid-list occurrences.

$$\text{RRF}(d) = \sum_{m=1}^{M} \frac{1}{\kappa + \text{rank}_m(d)} \tag{1}$$

We also consider **Average Ranking**, which directly averages ranks across retrievers; see Equation 6 for the formal definition.

### 2.2 SCORE-LEVEL AGGREGATION

Score-level aggregation operates one step deeper than ranking aggregation, operating on the real-valued similarity scores $s_m(q, d_i)$ between the query and documents. To ensure that the scores of different retrievers are in the same scale and range, the common practice (Bruch et al., 2023) is to first apply a normalization function $N_m$ – yielding $\tilde{s}_m(q, d_i)$ – and then aggregate across retrievers:

$$\tilde{s}_m(q, d_i) = N_m(s_m(q, d_i)), \qquad \text{Score}(q, d_i) = \frac{1}{M} \sum_{m=1}^{M} \tilde{s}_m(q, d_i). \tag{2}$$

---

[3]Milvus docs; Elasticsearch docs.

In this work, we evaluate two variants with different normalizations, **Score Aggregation (Min-Max)** and **Score Aggregation (SoftMax)**. See Appendix C for details.

**Tuned Variants.** More generally, the above methods can be viewed as a weighted aggregation, where the examples above are the uniform case, with each retriever assigned a weight $\alpha = 1/M$ (for two retrievers, $\alpha = 0.5$ for each). Given a development set, these weights can be fit (Bruch et al., 2023). Here, for $M = 2$, the two retrievers are assigned relative weights $\alpha$ and $1 - \alpha$, yielding the parameterized variants **Average Ranking -** *Tuned*, **RRF -** *Tuned*, **Score Aggregation (Min-Max) -** *Tuned*, and **Score Aggregation (SoftMax) -** *Tuned*. Details are in Appendix C.

## 2.3 GUIDED QUERY REFINEMENT

Representation-level information carries the richest potential for effective aggregation. Embedding-level projections that align representational spaces are used extensively in modern vision–language systems to combine visual and textual inputs (Radford et al., 2021; Jia et al., 2021; Li et al., 2021; 2022). At test time, however, operating directly on representations is hindered by heterogeneity: encoders may use a single vector or many vectors per document and query, and they operate within differing dimensionalities and scales. Thus, with strict latency and memory budgets and without access to supervision, aggregation at this level is not trivial.[4]

Our goal in this work is to exploit the rich information in query and document representations while remaining architecture-agnostic, lightweight, and practical. To this end, we propose *Guided Query Refinement* (*GQR*), a novel method for combining the outputs of two retrievers – a primary retriever $m_1$ and a complementary retriever $m_2$ (Algorithm 1, Figure 2). GQR refines $m_1$'s query representation based on the signal of $m_2$'s scores.

At inference time, given a user query $q$, an index search is run with each retriever to obtain its top-$K$ document list $\pi_m(q)$. The union of these lists, $\mathcal{C}(q) = \bigcup_{m=1}^{M} \pi_m(q)$, serves as the candidate pool of documents. For each retriever $m_j \in \{m_1, m_2\}$, we define a distribution over $\mathcal{C}(q)$ via a Softmax:

$$p_j(d_i \mid e_j^q) = \frac{\exp\big(s_j(q, d_i)\big)}{\sum_{k=1}^{|\mathcal{C}(q)|} \exp\big(s_j(q, d_k)\big)} \quad \text{for } i = 1, \ldots, |\mathcal{C}(q)|.$$

We denote the initial query embedding of $m_1$ by $z^{(0)} = e_1^q$ (GQR is applicable to both single and multi-vector embeddings), and we update it at each step $t$, $z^{(t)}$ for $T$ steps. At step $t$, the consensus distribution of $m_1$ and $m_2$ is defined as

$$p_{\text{avg}}^{(t)}(d) = \tfrac{1}{2}\Big(p_1\big(d \mid z^{(t)}\big) + p_2\big(d \mid e_2^q\big)\Big),$$

such that only $p_1$ depends on $t$ through $z^{(t)}$ and $p_2$ is fixed by $e_2^q$.

We minimize

$$\mathcal{L}^{(t)} = \text{KL}\big(p_{\text{avg}}^{(t)}(d) \,\|\, p_1(d \mid z^{(t)})\big).$$

Here, KL is the Kullback–Leibler divergence (Equation 12).

We apply a gradient step on the query representation with step size $\alpha$[5],

$$z^{(t+1)} = z^{(t)} - \alpha \, \nabla_z \mathcal{L}\big(z^{(t)}\big),$$

where $T$ and $\alpha$ are hyperparameters.

We then compute the final scores from retriever $m_1$,

$$s_1^{(T)}(q, d) = s_1\big(z^{(T)}, d\big) \quad \text{for } d \in \mathcal{C}(q).$$

Finally, we produce the list of retrieval results by sorting $\mathcal{C}(q)$ in descending order of $s_1^{(T)}(q, d)$, returning the first $K$ elements.

---

[4]Concatenating single-vector embeddings is feasible, yet under dot-product scoring this reduces to an unnormalized sum of separate scores, and does not enable interaction between the spaces.

[5]We define GQR with gradient descent for simplicity, but in practice we found Adam (Kingma & Ba, 2015) to perform better and use it as the optimizer.

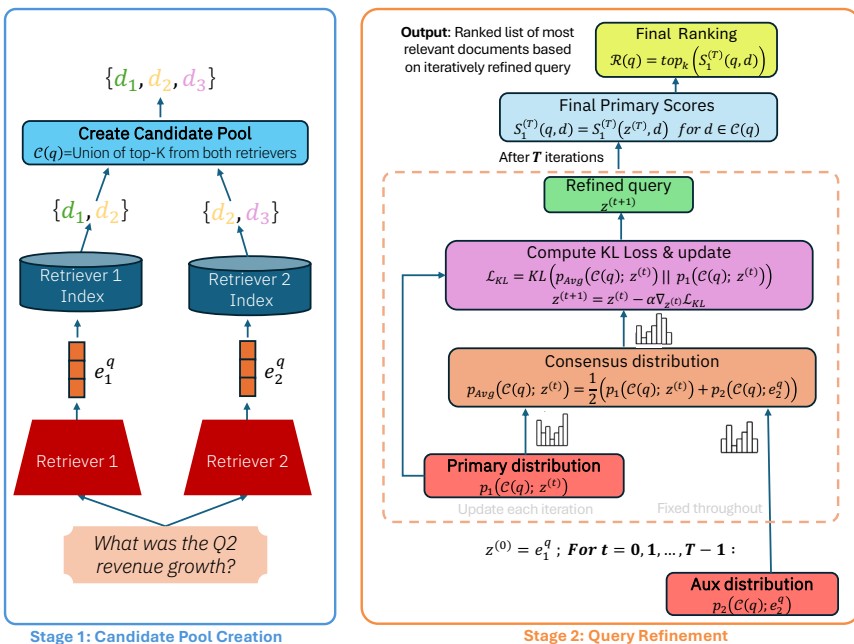

Figure 2: **Guided Query Refinement (GQR). Stage 1:** Two retrievers independently encode the query and retrieve top-$K$ documents, forming a candidate pool. **Stage 2:** The primary query embedding is iteratively refined ($z^{(t)}$) over $T$ iterations, by minimizing the KL divergence between a consensus distribution and the primary distribution.

## 2.4 GQR - MOTIVATION AND RATIONALE

Our approach is inspired by test time optimization methods that rely on pseudo-relevance feedback from a stronger cross-encoder (Yu et al., 2021; Sung et al., 2023; Gangi Reddy et al., 2025). Minimizing KL pushes $p_1(\cdot \mid z^{(t)})$ to place higher probability where the other distribution does, and to reduce probability where the other distribution is low. However, instead of relying on a heavy cross-encoder, here we utilize feedback from a lightweight bi-encoder whose performance can be on par or even weaker than the primary encoder, and thus the consensus distribution is set as $p_{\text{avg}}$.

Compared to a simple weighted average of scores, GQR operates by updating the query representation of $m_1$ rather than defining a new scoring rule. The complementary distribution $p_{\text{avg}}^{(t)}$ is used as a learning signal, and the gradient of $\mathcal{L}^{(t)}$ is computed through $s_1(z, d)$, so any change in the ranking must be achievable by moving the query within the primary model's embedding space. Intuitively, this allows feedback from $m_2$ to influence the query in a way that is constrained by $m_1$'s notion of similarity, since documents contribute to the update through their embeddings and their relative probabilities. In score-level fusion, each document's probability is pulled toward the secondary retriever's distribution by the same fixed amount, so the update is always a uniform weighted average. In GQR, different documents can shift by different magnitudes along a non-linear trajectory dictated by the geometry of that space. In cases where $m_2$ is weaker or misaligned, this setup can help incorporate its signal more softly, since it is filtered through $m_1$'s representation rather than directly overriding scores.

## 3 EXPERIMENTS

### 3.1 SETUP

**Task.** Visual document retrieval (Mathew et al., 2021b;a; Li et al., 2024; Zhu et al., 2022; Faysse et al., 2025) assumes a corpus of documents, that contain visual elements such as charts, images, and tables, and a set of document-grounded textual queries. The goal is to retrieve the most relevant

Table 1: NDCG@5 over ViDoRe 2, by primary and complementary models. Columns show scores by subset and the overall average. Deltas are absolute changes vs. the *No refinement* row within the same base.

| Primary Model | GQR complementary model | Avg | | Biomed Lectures | | Economics | | ESG Human | | ESG Full | |
|---|---|---|---|---|---|---|---|---|---|---|---|
| | | val | Δ | val | Δ | val | Δ | val | Δ | val | Δ |
| **Jina (text)** | | 53.4 | 0.0 | 48.6 | 0.0 | 51.4 | 0.0 | 59.5 | 0.0 | 54.1 | 0.0 |
| **Linq-Embed** | | 55.3 | 0.0 | 58.0 | 0.0 | 52.0 | 0.0 | 58.8 | 0.0 | 52.4 | 0.0 |
| **Qwen3** | | 46.8 | 0.0 | 54.0 | 0.0 | 44.6 | 0.0 | 50.2 | 0.0 | 38.3 | 0.0 |
| **Colnomic-7B** | | | | | | | | | | | |
| | No refinement | 60.3 | 0.0 | 64.3 | 0.0 | 54.4 | 0.0 | 68.2 | 0.0 | 54.1 | 0.0 |
| | Jina (text) | **63.1** | ↑+2.8 | 64.7 | ↑+0.4 | **57.0** | ↑+2.6 | **70.3** | ↑+2.1 | 60.2 | ↑+6.1 |
| | Linq-Embed | 62.8 | ↑+2.5 | **65.4** | ↑+1.1 | 56.7 | ↑+2.3 | 67.7 | ↓1.0 | **61.2** | ↑+7.1 |
| | Qwen3 | 61.0 | ↑+0.7 | 61.9 | ↓2.4 | 54.3 | ↓0.1 | 70.2 | ↑+2.0 | 57.5 | ↑+3.4 |
| **Jina (vision)** | | | | | | | | | | | |
| | No refinement | 57.2 | 0.0 | 61.6 | 0.0 | 53.5 | 0.0 | 61.7 | 0.0 | 52.0 | 0.0 |
| | Jina (text) | 60.7 | ↑+3.5 | 61.7 | ↑+0.1 | 55.3 | ↑+1.8 | 66.9 | ↑+5.2 | **58.8** | ↑+6.8 |
| | Linq-Embed | **61.2** | ↑+4.0 | **64.7** | ↑+3.1 | **57.2** | ↑+3.7 | 65.7 | ↑+4.0 | 57.1 | ↑+5.1 |
| | Qwen3 | 59.8 | ↑+2.6 | 63.2 | ↑+1.6 | 53.6 | ↑+0.1 | **67.8** | ↑+6.1 | 54.4 | ↑+2.4 |
| **Llama-Nemo** | | | | | | | | | | | |
| | No refinement | 63.0 | 0.0 | 63.7 | 0.0 | 56.8 | 0.0 | 74.5 | 0.0 | 56.9 | 0.0 |
| | Jina (text) | 64.2 | ↑+1.2 | 64.5 | ↑+0.8 | **57.6** | ↑+0.8 | 74.2 | ↓0.3 | 60.4 | ↑+3.5 |
| | Linq-Embed | **65.2** | ↑+2.2 | **66.4** | ↑+2.7 | 56.8 | 0.0 | **74.6** | ↑+0.1 | **62.8** | ↑+5.9 |
| | Qwen3 | 63.3 | ↑+0.3 | 65.0 | ↑+1.3 | 55.4 | ↓1.4 | 74.1 | ↓0.4 | 58.7 | ↑+1.8 |

documents for each query. We conduct experiments on *ViDoRe 1* (Faysse et al., 2025), *ViDoRe 2* (Macé et al., 2025) and the *ViDoRe 3* benchmarks, which are established benchmarks for this task. Corpus documents are embedded by encoder models, either directly from page images, or following ingestion of document pages into text (see Appendix D).

**Models.** We evaluate a diverse pool of multimodal and textual state-of-the-art retrieval models. The Colpali-based set includes three encoders: COLNOMIC-EMBED-MULTIMODAL-7B (Team, 2025b), JINA-EMBEDDINGS-V4 (Günther et al., 2025), and LLAMA-NEMORETRIEVER-COLEMBED-3B (Xu et al., 2025). The set of text models includes LINQ-EMBED-MISTRAL (Choi et al., 2024) and QWEN3-EMBEDDING-4B (Zhang et al., 2025), as well as JINA-EMBEDDINGS-V4 in its multi-vector textual configuration. This yields 3 text-based models and 3 image-based models in total. See Table 4 for details on the models. For the ViDoRe 3 benchmark, we excluded LLAMA-NEMORETRIEVER-COLEMBED-3B due to its high computational overhead, which makes evaluation impractical over this larger benchmark.

**Metrics and Evaluation.** We use NDCG@5 as the primary metric for our evaluations (we also report Recall@5 in Appendix F.) For each *ColPali*-based vision-centric model, we test each of the text-centric models as the complementary retriever used for GQR. This yields 9 GQR pairs in total for ViDoRe 1 and ViDoRe 2 and 6 for ViDoRe 3, 3 per *ColPali*-based model. We also evaluate 4 different hybrid methods for each vision-text model pair. See Appendix D for technical details about tuning the hyperparameters for ViDoRe 1 and ViDoRe 2.

To simulate real production use cases, we evaluate both tuned and zero-shot settings. We begin with the tuned setting on ViDoRe 1 and ViDoRe 2, and then use the findings from ViDoRe 2 to guide a zero-shot evaluation on ViDoRe 3.

### 3.2 COMPARISON TO SINGLE-RETRIEVER PIPELINES

**Results with Hyperparameter Tuning.** Table 1 reports GQR against the corresponding models on ViDoRe 2. The first block lists the text-only models, which average between 46.8 for Qwen and 55.3 for Linq. In each subsequent block, a ColPali-based retriever is fixed, and we show its score alongside the GQR variants, with deltas computed relative to the primary retriever. For Colnomic-

Table 2: NDCG@5 over ViDoRe 3, by primary and complementary models. Columns show subset scores, average, and deltas relative to the no-refinement baseline within each primary model.

| Primary Model | GQR complementary model | Avg | | cs | | energy | | fin_en | | fin_fr | | hr | | industrial | | pharma | | physics | |
|---|---|---|---|---|---|---|---|---|---|---|---|---|---|---|---|---|---|---|---|
| | | val | Δ | val | Δ | val | Δ | val | Δ | val | Δ | val | Δ | val | Δ | val | Δ | val | Δ |
| **Jina (text)** | | 51.6 | 0.0 | 66.6 | 0.0 | 60.2 | 0.0 | 54.0 | 0.0 | 40.1 | 0.0 | 54.5 | 0.0 | 42.2 | 0.0 | 55.3 | 0.0 | 40.5 | 0.0 |
| **Linq-Embed** | | 44.2 | 0.0 | 64.6 | 0.0 | 48.5 | 0.0 | 40.1 | 0.0 | 28.7 | 0.0 | 42.6 | 0.0 | 30.1 | 0.0 | 57.7 | 0.0 | 41.4 | 0.0 |
| **Qwen3** | | 46.0 | 0.0 | 67.6 | 0.0 | 50.1 | 0.0 | 38.9 | 0.0 | 27.3 | 0.0 | 44.6 | 0.0 | 37.8 | 0.0 | 58.1 | 0.0 | 43.9 | 0.0 |
| **Colnomic-7B** | | | | | | | | | | | | | | | | | | | |
| | No refinement | 55.7 | 0.0 | 74.0 | 0.0 | 62.3 | 0.0 | 54.2 | 0.0 | 42.7 | 0.0 | 57.1 | 0.0 | 49.5 | 0.0 | 62.0 | 0.0 | 44.1 | 0.0 |
| | Jina (text) | 57.4 | ↑+1.7 | 74.8 | ↑+0.8 | 64.1 | ↑+1.8 | 58.6 | ↑+4.4 | 44.6 | ↑+1.9 | 59.4 | ↑+2.3 | 50.8 | ↑+1.3 | 62.8 | ↑+0.8 | 44.5 | ↑+0.4 |
| | Linq-Embed | 57.1 | ↑+1.4 | 75.1 | ↑+1.1 | 63.4 | ↑+1.1 | 56.9 | ↑+2.7 | 43.9 | ↑+1.2 | 58.1 | ↑+1.0 | 50.8 | ↑+1.3 | 63.4 | ↑+1.4 | 45.7 | ↑+1.6 |
| | Qwen3 | 56.7 | ↑+1.0 | 75.2 | ↑+1.2 | 63.1 | ↑+0.8 | 56.0 | ↑+1.8 | 42.6 | ↓-0.1 | 58.0 | ↑+0.9 | 50.5 | ↑+1.0 | 63.1 | ↑+1.1 | 45.4 | ↑+1.3 |
| **Jina (vision)** | | | | | | | | | | | | | | | | | | | |
| | No refinement | 53.4 | 0.0 | 69.2 | 0.0 | 59.4 | 0.0 | 52.4 | 0.0 | 39.8 | 0.0 | 54.7 | 0.0 | 47.8 | 0.0 | 61.2 | 0.0 | 43.0 | 0.0 |
| | Jina (text) | 55.1 | ↑+1.7 | 70.2 | ↑+1.0 | 61.5 | ↑+2.1 | 56.3 | ↑+3.9 | 42.6 | ↑+2.8 | 57.1 | ↑+2.4 | 48.1 | ↑+0.3 | 61.6 | ↑+0.4 | 43.5 | ↑+0.5 |
| | Linq-Embed | 55.4 | ↑+2.0 | 72.0 | ↑+2.8 | 61.2 | ↑+1.8 | 54.9 | ↑+2.5 | 41.5 | ↑+1.7 | 56.6 | ↑+1.9 | 48.6 | ↑+0.8 | 63.8 | ↑+2.6 | 45.1 | ↑+2.1 |
| | Qwen3 | 55.0 | ↑+1.6 | 72.3 | ↑+3.1 | 60.4 | ↑+1.0 | 54.1 | ↑+1.7 | 39.7 | ↓-0.1 | 55.9 | ↑+1.2 | 49.3 | ↑+1.5 | 63.1 | ↑+1.9 | 45.6 | ↑+2.6 |

Table 3: Percentage gain in NDCG@5 of hybrid retrieval over the primary retriever on ViDoRe 2. Each cell depicts the average gain over 9 retriever pairs (3 multimodal base retrievers and 3 text retrievers).

| Method | Avg | Std | Per-Subset Gain | | | |
|---|---|---|---|---|---|---|
| | | | Biomed Lectures | Economics | ESG Human | ESG Full |
| Average Ranking | ↓-3.0% | 2.5 | ↓-6.6% | ↑+0.5% | ↓-6.4% | ↑+0.4% |
| RRF | ↓-2.8% | 2.6 | ↓-6.5% | ↑+0.3% | ↓-6.2% | ↑+1.3% |
| Score Aggregation (Min-Max) | ↑+0.4% | 2.6 | ↓-3.0% | ↑+1.7% | ↓-1.8% | ↑+4.7% |
| Score Aggregation (Softmax) | ↑+1.5% | 1.9 | ↓-0.9% | ↑+0.8% | ↑+0.6% | ↑+5.4% |
| Average Ranking (Tuned) | ↓-0.3% | 1.8 | ↓-2.9% | ↑+1.3% | ↓-2.8% | ↑+3.2% |
| RRF (Tuned) | ↓-0.1% | 1.9 | ↓-2.9% | ↑+1.2% | ↓-4.0% | ↑+5.5% |
| Score Aggregation (Min-Max, Tuned) | ↑+3.4% | 2.1 | ↑+1.2% | ↑+2.9% | ↑+2.9% | ↑+6.7% |
| Score Aggregation (Softmax, Tuned) | ↑+2.6% | 2.3 | ↑+0.3% | ↑+0.9% | ↑+2.2% | ↑+7.1% |
| *Guided Query Refinement (GQR)* | ↑+3.9% | 1.9 | ↑+1.5% | ↑+2.0% | ↑+3.3% | ↑+8.7% |

7B, the average score rises from 60.3 to 63.1 with query refinement from Jina (text) $(+2.8)$ and to 62.8 with Linq-Embed $(+2.5)$. For Llama-Nemo, the strongest model on ViDoRe 2 to date, GQR improves the average from 63.0 to 65.2 with Linq-Embed $(+2.2)$. Notably, the text models clearly underperform the *ColPali*-based retrievers, yet with GQR the complementary signal they provide boosts performance. This is clearest in the Llama-Nemo versus Qwen3 setting, where GQR delivers a small gain of ↑+0.3, despite a 16.2 point gap in base NDCG@5 (Llama-Nemo 63.0 vs. Qwen 46.8). Subsets where GQR harms performance are rare for Colnomic-7B and Llama-Nemo, absent for Jina (vision), and on average across the benchmark GQR consistently improves retrieval quality. We observe similar patterns for the Recall@5 metric (Table 12). On ViDoRe 1 (Table 7 in the Appendix) GQR is generally on par with the base models, yet the benchmark clearly suffers from saturation, with many subset scores reaching 90 or higher.

**Results - Zero Shot.** For ViDoRe 3, we use a single hyperparameter configuration across all subsets and model pairs, a choice informed by our hyperparameter investigation on top of ViDoRe 2 in subsection 4.1. Table 2 shows that GQR improves performance across all model pairs and benchmark subsets, consistently outperforming the no-refinement baseline. For example, on Colnomic-7B it raises the average NDCG@5 from 55.7 to 57.4 with Jina as the complementary model (a gain of $+1.7$), and delivers per-subset gains as high as $+4.4$ on fin_en. Similar improvements appear for Jina (vision), where GQR increases the average from 53.4 to 55.4 with Linq-Embed $(+2.0)$.

### 3.3 COMPARISON TO HYBRID RETRIEVAL PIPELINES

Table 3 depicts the aggregated performance of the different hybrid retrieval methods on ViDoRe 2, presented as the average percentage gain relative to the base retriever over all pairs (Table 6 in the appendix lists the average absolute values). The *Std* column captures the standard deviation of performance across the pairs for each method. It shows that the ranking aggregation methods (RRF,

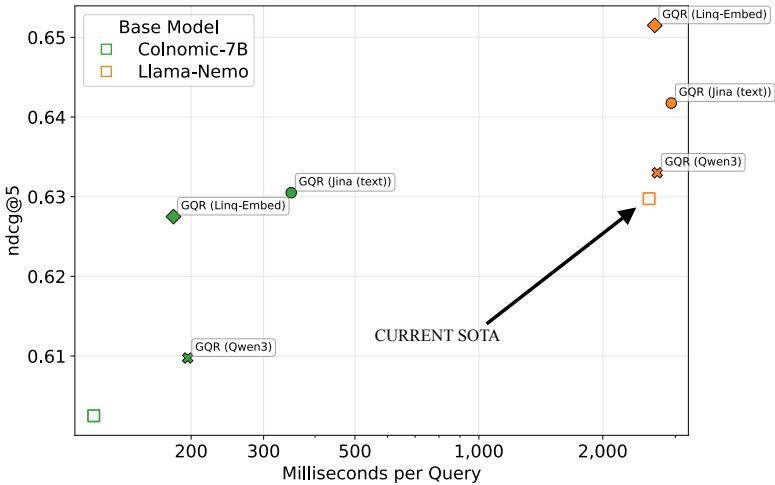

Figure 3: Latency–quality tradeoff in online querying. The $x$ axis is runtime in milliseconds for a single query, on a log scale, and the $y$ axis is the average evaluation score (NDCG@5). Empty squares indicating the primary retriever alone (without applying GQR).

Average Ranking) generally lead to a deterioration in performance. All aggregation methods benefit from parameter tuning, with tuned score aggregation methods achieving consistent gains over the base retriever. Table 5 in Appendix F shows the results in zero-shot settings on ViDoRe 3 (see Figure 7 for how the baseline zero-shot configuration is chosen).

Across the benchmarks, GQR outperforms all other hybrid retrieval variants, with the largest average gains over the base multimodal retriever, while also emerging as the most stable positive method with a low standard deviation across pairs.

### 3.4 Results - Efficiency

**Online Querying Latency.** To characterize the test-time costs of GQR and *ColPali*-based retrievers, we ran latency measurements on a single NVIDIA A100 GPU, measuring document retrieval for randomly sampled queries from each ViDoRe 2 subset (averaging across 100 runs). Figure 3 depicts the quality-latency trade-offs of the different retrievers, where each point on the graph represents either a base retriever or a retriever with GQR (Appendix F shows full values). The plot illustrates the substantial costs of the strongest base model – Llama-Nemo (orange square, right) – which attains NDCG@5 = 62.9 at a cost of 2,591 ms per query. Notably, our Colnomic GQR hybrid, with Linq as the refinement model (green diamond, left), reaches NDCG@5 = 62.7 at 181 ms ($\approx 14\times$ faster), and the Colnomic hybrid with Jina (green circle) attains NDCG@5 = 63.0 at 350 ms ($\approx 7\times$ faster), surpassing Nemo. Across base models, applying GQR increases latency by small relative measures with large gains in performance, shifting the Pareto frontier left and upwards. Appendix F shows that GQR is also on the index-storage Pareto frontier.

## 4 Analysis

### 4.1 Hyperparameter investigation

GQR relies on two hyperparameters - the learning rate $\alpha$ and the number of optimization steps $T$. We evaluate performance over the Cartesian product $\{1 \times 10^{-5}, 5 \times 10^{-5}, 10^{-4}, 5 \times 10^{-4}, 10^{-3}, 5 \times 10^{-3}\} \times \{5, 10, 15, \ldots, 90, 95, 100\}$ of $\alpha$ and $T$. We run this analysis over 6 model pairs, excluding LLAMA-NEMORETRIEVER-COLEMBED-3B due to its computational overhead.

Figure 4 depicts GQR performance on ViDoRe 2 for each combination of $\alpha$ and $T$, averaged over the 6 model pairs. It shows that high learning rates ($10^{-3}$, $5 \times 10^{-4}$) are compatible with small $T$ values (10-25), where beyond that performance starts decreasing. Medium learning rates ($10^{-4}$, $5 \times 10^{-5}$), show consistent improvement and stable optimization, yet require more steps (75-100) to reach

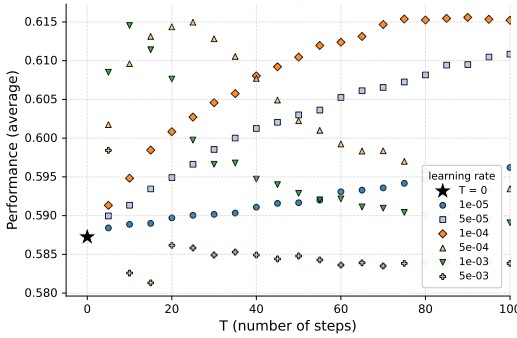 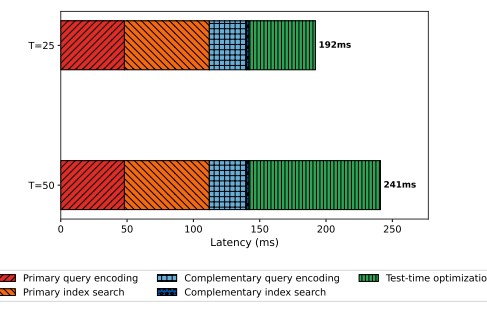

Figure 4: Hyperparameter sweep over GQR's learning rate $\alpha$ and optimization steps $T$, averaged over six model pairs on ViDoRe 2.

Figure 5: Online latency breakdown of GQR for $T = 25$ and $T = 50$.

higher performance. The smallest ($10^{-5}$) and largest ($5 \times 10^{-3}$) learning rates are suboptimal, where the latter even results in performance degradation relative to the primary retriever.

The results capture a tradeoff between latency and stability. Higher learning rates can provide a performance boost faster, but might deteriorate quickly past a certain $T$ value, while medium learning rates are more stable yet require more online optimization steps. Overall, three configurations emerge as the most promising for GQR: (1) $\alpha = 10^{-3}$ with $5 < T < 15$, (2) $\alpha = 5 \times 10^{-4}$ with $15 < T < 30$, and (3) $\alpha = 10^{-4}$ with $50 < T < 75$. We leave it to future work to explore methods for optimizing this tradeoff, potentially by incorporating learning-rate scheduling procedures.

## 4.2 ONLINE LATENCY BREAKDOWN

Figure 5 describes the online latency breakdown of GQR, for $T = 25$ and $T = 50$. The primary model used is COLNOMIC-EMBED-MULTIMODAL-7B and the complementary is LINQ-EMBED-MISTRAL. Latencies are collected with an A100 gpu over the ViDoRe 2 benchmark.

The test-time optimization accounts for 50 ms of a 192 ms total latency for $T = 25$ and 99 ms of a 241 ms total for $T = 50$, averaging roughly 2 ms per optimization step. The procedure is shown here in a sequential form, although with sufficient GPU memory the query encoding and index search for the two retrievers can be parallelized, reducing overall latency.

## 4.3 RERANKER COMPARISON

Cross-encoder rerankers are widely used in retrieval pipelines, applying full query to document interaction via self-attention to the top-K documents. Similarly to GQR, rerankers operate at test time on a candidate pool returned by a bi-encoder, and thus provide a natural point of comparison. We evaluate GQR against *lightonai/MonoQwen2-VL-v0.1*, an open-weights multimodal reranker (Chaffin & Lac, 2024). Figure 6 illustrates the latency-performance characteristics of GQR against reranking the top-5 (on the left) and the top-10 (on the right) candidates returned by each retriever. GQR is run with a default $K = 10$ configuration and with Linq-Embed as the refinement model. The left plot illustrates that GQR outperforms a top-5 reranking pipeline on both the latency and performance axes. Against the top-10 reranking pipelines (right), GQR achieves close performance while being $21\times$ faster for Colnomic, $16\times$ faster for Jina (vision), and $2\times$ faster for Nemo. Across both comparisons, GQR remains on the Pareto front and indicates attractive latency performance trade-offs. We report the full results in Table 11.

## 4.4 GQR DESIGN CHOICES

**Extra search stage.** Prior works on test-time query optimization often run a second index search with the optimized query $z^{(T)}$, aiming to retrieve documents beyond the initial top-$K$ and increase recall (Sung et al., 2023; Gangi Reddy et al., 2025). This extra pass adds latency as it repeats index traversal and candidate generation. We test this modification to GQR where we perform an

additional search with the optimized query over the full index, and observe no improvement in performance (Table 16). This suggests that the effects of GQR in our setting are largely confined to the original pool $\mathcal{C}(q)$ of candidate documents.

**Choice of objective.** The GQR optimization process minimizes the KL-divergence between the distribution of the primary retriever and a consensus distribution of the two retrievers, $\mathrm{KL}(p_{\mathrm{avg}} \| p_1)$. We additionally test two other loss functions: The Jensen-Shannon divergence, $\mathrm{JS}(p_2 \| p_1) = \frac{1}{2}\mathrm{KL}(p_2 \| p_{\mathrm{avg}}) + \frac{1}{2}\mathrm{KL}(p_1 \| p_{\mathrm{avg}})$ and KL with the target distribution $p_2$, i.e., $\mathrm{KL}(p_2 \| p_1)$. We find (Table 18) that these GQR variants generally perform similarly well, indicating that GQR applies across different loss formulations of the two distributions.

We conduct additional experiments on GQR design choices in Appendix E.

## 5 RELATED WORKS

**Multimodal Retrieval.** Recent advances in visual document retrieval have been dominated by late-interaction, multi-vector architectures. This paradigm, first introduced to the multimodal domain by ColPali (Faysse et al., 2025), adapts the ColBERT framework (Khattab & Zaharia, 2020) by treating image patches as visual tokens that interact with textual query tokens via MaxSim operations. Subsequent models built on this foundation – such as Llama-NemoRetriever-ColEmbed (Xu et al., 2025) and ColNomic-Embed-Multimodal (Team, 2025b) – have established state-of-the-art performance by capturing fine-grained interactions that are lost in single-vector representations. However, this performance comes at a significant cost: to achieve their results, these models rely on massively scaled representations, leading to substantial latency and storage overheads that can hinder practical deployment. This pressing trade-off between performance and efficiency motivates our work. Instead of pursuing ever-larger monolithic models, we investigate an alternative direction: enhancing these powerful vision-centric retrievers by fusing their signal with a complementary lightweight text-based encoder at test time.

**Hybrid Search.** Numerous works apply hybrid retrieval in the context of combining dense semantic text retrieval with sparse lexical representations (Karpukhin et al., 2020; Kuzi et al., 2020; Luan et al., 2021; Chen et al., 2022). Bruch et al. (2023) conduct a theoretical and empirical analysis of the different ways to perform such dense-sparse fusions. Specifically, they compare RRF (Cormack et al., 2009) to score-based fusion, and analyze the sensitivity to the choice of tuned weights and normalizations. Hsu & Tzeng (2025) propose to set the score-fusion weights dynamically for each query at test time, based on (costly) feedback from an LLM judge. In contrast, GQR departs from these approaches by operating at test time on the representation level of the primary retriever.

**Test-time query refinement.** Prior work optimizes query representations during inference in text-only setups using pseudo-relevance feedback, often distilling from a cross-encoder re-ranker to a single-vector dense retriever (Yu et al., 2021; Sung et al., 2023; Gangi Reddy et al., 2025). Cross-encoders provide rich interactions, but incur substantial test-time cost. Here we replace the cross-encoder with a complementary bi-encoder (possibly of a different modality), which preserves low latency while still providing a strong guidance signal.

## 6 CONCLUSION

In this work, we introduced Guided Query Refinement (GQR), a novel test-time hybrid retrieval method that refines the query representations of a primary retriever using signals from a complementary one. Unlike traditional hybrid techniques that operate on rankings or scores, GQR leverages representation-level interactions while maintaining efficiency and modularity.

Through extensive experiments on the ViDoRe benchmarks, we demonstrated that GQR consistently improves retrieval performance across diverse model pairs, pushing ColPali-based retrievers to the latency–memory Pareto frontier. Our findings highlight that even weaker retrievers can provide valuable complementary guidance, underscoring the potential for resource-efficient retrieval systems in multimodal large-scale settings.

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

## A  BACKGROUND

Neural Information Retrieval represents a fundamental shift from traditional lexical matching methods like BM25 (Robertson et al., 1995), TF-IDF (Salton & Buckley, 1988), and other term-based approaches (Zhai & Lafferty, 2017). Unlike these sparse retrieval methods that rely on exact term matches and statistical properties, neural approaches learn dense semantic representations that capture conceptual similarity, enabling retrieval based on meaning rather than just shared vocabulary.

DENSE ENCODERS have transformed information retrieval by learning semantic representations of queries and passages in a shared embedding space. These models typically produce a single dense vector representation for each passage and query, enabling efficient similarity computation through operations like cosine similarity or inner product. Early work by Karpukhin et al. (2020) introduced Dense Passage Retrieval (DPR), demonstrating that dense representations could outperform traditional sparse methods like BM25 for open-domain question answering. For query, $q$ and passage $p$, The similarity score $S$ is defined as:

$$S(q,p) = \frac{q \cdot p}{|q||p|} \tag{3}$$

The BERT-based bi-encoder architecture, where queries and passages are independently encoded using separate or shared BERT models, became the foundational paradigm for neural retrieval. This framework has shaped the field, with numerous dense retrieval models building upon it (Reimers & Gurevych, 2019; Xiong et al., 2020; Qu et al., 2020; Izacard et al., 2021). The bi-encoder design enables pre-computation of passage embeddings, making real-time retrieval feasible at scale. Recently, advanced models such as Linq-Embed-Mistral (Choi et al., 2024) and Qwen3-Embedding (Zhang et al., 2025) extend this paradigm by leveraging larger language models as the underlying encoder to create more powerful dense representations.

LATE-INTERACTION MODELS compute query-document similarity through more fine-grained interaction mechanisms. Rather than compressing all information into a single vector, these models preserve individual token embeddings for both queries and passages. ColBERT (Khattab & Zaharia, 2020) pioneered this approach with its MaxSim operation, which computes the maximum similarity between each query token and all passage tokens, then aggregates these scores. The MaxSim equation, as defined in ColBERT (see eq. 4)), finds for query token embedding $q_i$ the maximum similarity (dot product) with any passage token embedding $p_j$. These maximum scores are then summed up to get the final relevance score.

$$S(q, p) = \sum_{i=1}^{|Q|} \max_{j=1}^{|P|} q_i \cdot p_j^T \tag{4}$$

This approach balances effectiveness and efficiency, as passage representations can still be pre-computed and indexed while the token-level matching captures finer details than single-vector approaches. Subsequent work improved efficiency and retrieval quality through various optimizations (Santhanam et al., 2021; 2022).

CONTRASTIVE TRAINING forms the backbone of modern retrieval model optimization. These methods learn representations by pulling positive query-passage pairs closer while pushing negative pairs apart in the embedding space. The InfoNCE loss (Oord et al., 2018), which maximizes the similarity of positive pairs relative to negative ones through a softmax-like normalization, has become the dominant training objective across retrieval architectures.

$$\mathcal{L} = -\log \frac{\exp(\text{sim}(q, p^+)/\tau)}{\exp(\text{sim}(q, p^+)/\tau) + \sum_{i=1}^{k} \exp(\text{sim}(q, p_i^-)/\tau)} \tag{5}$$

CROSS-ENCODER RE-RANKERS represent a different paradigm where query and passage are jointly encoded, enabling deep self-attention between their representations. Unlike bi-encoders that independently encode queries and passages, cross-encoders process query-passage pairs through a single model, allowing full attention across all tokens. This joint encoding is computationally expensive, making it impractical for first-stage retrieval over large corpora. However, cross-encoders excel as re-rankers, refining the top-K results from efficient first-stage retrievers. Examples include MonoBERT (Nogueira et al., 2019) for text retrieval and Chaffin & Lac (2024); Wasserman et al. (2025) for visual document retrieval.

## A.1 VISUAL DOCUMENT RETRIEVAL

VERBALIZATION-BASED METHODS were the dominant approach before the advent of end-to-end vision models. These pipelines convert visual documents into text through various techniques: traditional Optical Character Recognition (OCR) tools like docling (Auer et al., 2024) extract printed text, while Vision-Language Models (VLMs) can generate textual descriptions of visual elements such as charts, diagrams, and infographics. After verbalization, these methods apply standard text retrieval techniques to the extracted content. While verbalization-based approaches can leverage powerful text-only retrieval models, they inherently lose spatial relationships and visual context during the text extraction process.

COLPALI ARCHITECTURES introduced a new approach to visual document retrieval by directly encoding document images without intermediate text extraction. These VLM-based embedding

Table 4: Different retrieval models evaluated in our work. The modality column describes the representation used for the documents – page-level text, or page-level images.

| Model | Size | # Vectors per Page | Token Dim | # Floats per Page | Storage per 1M Docs (GB) | Page Modality |
|---|---|---|---|---|---|---|
| LINQ-EMBED-MISTRAL | 7.1B | 1 | 4096 | 4096 | 7.63 | Text |
| QWEN3-EMBEDDING-4B | 4B | 1 | 2560 | 2560 | 4.77 | Text |
| JINA-EMBEDDINGS-V4 (Text) | 3.8B | – | 128 | – | – | Text |
| JINA-EMBEDDINGS-V4 (Image) | 3.8B | 767 | 128 | 98176 | 182.87 | Image |
| COLNOMIC-EMBED-MULTIMODAL-7B | 7B | 767 | 128 | 98176 | 182.87 | Image |
| LLAMA-NEMORETRIEVER-COLEMBED-3B | 4.4B | 1802 | 3072 | 5535744 | 10311.13 | Image |

models transform pre-trained generative Vision-Language Models (such as PaliGemma, Beyer et al., 2024 and Qwen-VL, Wang et al., 2024) into multi-vector embedding models optimized for retrieval.

ColPali (Faysse et al., 2025) was the first model to introduce this approach, building upon PaliGemma (Beyer et al., 2024) and adapting the late-interaction framework to vision-language models by treating image patches as visual tokens that interact with textual query tokens through MaxSim operations. ColPali provides native text-query support due to its VLM-based design. Queries remain text, are encoded by the model's language tower, and are matched directly against visual page/passage tokens, eliminating any OCR at query time. Training is contrastive, typically InfoNCE-style with in-batch/hard negatives to align query tokens with relevant visual tokens. This architecture preserves spatial layout information and visual features that verbalization-based methods discard. Following ColPali's success, subsequent models have adopted and extended this ColPali paradigm by leveraging different base VLMs: Llama-NemoRetriever-ColEmbed (Xu et al., 2025), Colnomic-Embed-Multimodal (Team, 2025b), Granite-Vision-Embedding (Team, 2025a) and Jina-Embeddings-v4 (Günther et al., 2025).

## B  MODEL INFORMATION

Table 4 depicts the details of the models evaluated in our work. Storage assumes a 16-bit representation and follows official reports (Xu et al., 2025). JINA-EMBEDDINGS-V4 supports both text and image document representations, single-vector and multi-vector retrieval, and flexible embedding sizes via a Matryoshka scheme (Kusupati et al., 2022). In its multi-vector textual configuration, the number of vectors per page for Jina varies between pages.

## C  ADDITIONAL DEFINITIONS

**Average Ranking**  Computes the average rank across retrievers.

$$\text{AvgRank}(d) = \frac{1}{M} \sum_{m=1}^{M} \text{rank}_m(d) \tag{6}$$

**Min-Max normalization**

$$\tilde{s}_m(q, d_i) = \frac{s_m(q, d_i) - \min_{d_j \in \pi_m(q)} s_m(q, d_j)}{\max_{d_j \in \pi_m(q)} s_m(q, d_j) - \min_{d_j \in \pi_m(q)} s_m(q, d_j) + \varepsilon}, \tag{7}$$

with a small $\varepsilon$ for numerical stability.

**Softmax normalization**

$$\tilde{s}_m(q, d_i) = \frac{\exp(s_m(q, d_i))}{\sum_{d_j \in \pi_m(q)} \exp(s_m(q, d_j))}. \tag{8}$$

If $d_i \notin \pi_m(q)$ we set $\tilde{s}_m(q, d_i) = 0$

**Average Ranking - Tuned (for M=2)**

$$\text{AvgRank}(d) = \alpha \ rank_{m_1}(d) + (1 - \alpha) \ rank_{m_2}(d) \tag{9}$$

**RRF - Tuned (for M=2)**

$$\text{RRF}(d) = \frac{2\alpha}{\kappa + \text{rank}_{m_1}(d)} + \frac{2(1 - \alpha)}{\kappa + \text{rank}_{m_2}(d)} \tag{10}$$

**Score aggregation - Tuned (for M=2)**

$$\text{Score}(q, d_i) = \alpha \ \tilde{s}_{m_1}(q, d_i) + (1 - \alpha) \ \tilde{s}_{m_2}(q, d_i) \tag{11}$$

**KL divergence.**

$$\text{KL}(P\|Q) = \sum_{d \in \mathcal{C}(q)} P(d) \ \log\frac{P(d)}{Q(d)}, \tag{12}$$

where $P$ and $Q$ are distributions over $\mathcal{C}(q)$.

## D  IMPLEMENTATION DETAILS

**Offline indexing.**  For each model and dataset, we construct an offline index of page-level document representations. For ColPali-based models, multi-page documents are rendered as page images and encoded directly. For text-only embedding models, for the ViDoRe 1 and ViDoRe 2 datasets an ingestion pipeline converts each page to text. We use Docling[6](Team, 2024) to ingest the images. Docling is an open library providing OCR capabilities combined with document layout analysis, allowing us to recover page content via simple function calls. The resulting text is stored alongside the page images without any chunking, ensuring consistent alignment between visual and textual page representations across the datasets. We run the document converter from Docling v2.34 using default parameters. For ViDoRe 3, we use the official markdown of the benchmark as the text input for the embedding models.

For JINA-EMBEDDINGS-V4 (Günther et al., 2025), a single model accepts both image and text input documents. We thus use it in both a vision configuration and in a text configuration, denoted *Jina (Vision)* and *Jina (Text)*, both running in a multi-vector setting. *Jina (Text)* is thus used to test the applicability of GQR where a multi-vector architecture is used as the the GQR complementary encoder.

**Hyperparameters**  For RRF we set $\kappa = 60$, a common default (Chen et al., 2022; Cormack et al., 2009). For tuning the weighted hybrid baselines and GQR, we follow previous works (Sung et al., 2023; Gangi Reddy et al., 2025; Bruch et al., 2023) and rely on an in-domain development set of queries. We reserve 10% of each subset and tune the hyperparameters, $T$, and $\alpha$ for each, selecting by development set NDCG@5 performance. The splits are fixed for all experiments. For the weighted hybrid variants, the weight $\alpha$ is tuned over $\{0.1, 0.2, \ldots, 0.9\}$ for each subset. For GQR, the learning rates are $\{1 \times 10^{-5}, 5 \times 10^{-5}, 10^{-4}, 5 \times 10^{-4}, 10^{-3}, 5 \times 10^{-3}\}$, and we consider $T \in \{10, 25, 50\}$. We report the selected hyperparameters for all methods in Tables 20-25. We opted for Adam (Kingma & Ba, 2015) as the optimizer for GQR, based on preliminary experiments comparing different optimizer choices. We use $K = 10$ across all retrievers and methods.

## E  ADDITIONAL DESIGN CHOICES FOR GQR

**Candidate pool policy.**  In our implementation the pool $\mathcal{C}(q)$ of candidate documents used for query refinement is a union of the top-K documents from the primary and complementary retrievers. One alternative is to opt for a reranker-like setup, where the only documents considered are those initially returned in the top-K of the primary retriever. Our analysis shows (Table 17) that this modification does not have a consistent positive or negative effect compared to a union pool of candidates. While the complementary dense retriever's index search is relatively quick, this configuration can be adopted in sensitive deployments of GQR for additional latency gains.

---

[6]https://docling-project.github.io/docling/

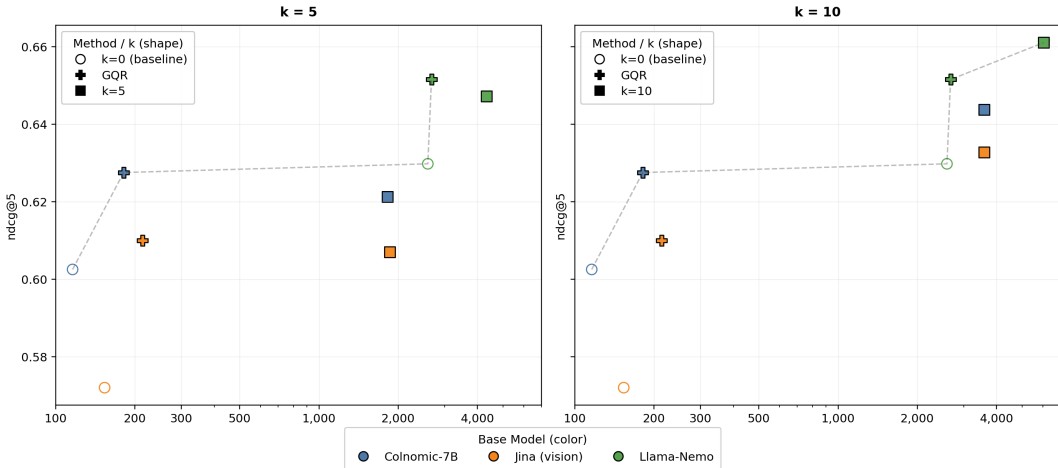

Figure 6: Latency–quality tradeoff in online querying. The $x$ axis is runtime in milliseconds for a single query, on a log scale, and the $y$ axis is the average evaluation score (NDCG@5). Marker color encodes the primary model; marker shape encodes the usage of GQR or reranking with top 5 (left) or top 10 (right) candidates. The dashed lines represent the Pareto frontier.

---

**Algorithm 1** *Guided Query Refinement (GQR)*

---

**Require:** Query $q$, primary encoder $m_1$, complementary encoder $m_2$, iterations $T$, step size $\alpha$, top-$K$ value $K$

1: $z^{(0)} \leftarrow e_1^q$      ▷ Initialize the primary encoder's query embedding
2: $\mathcal{C}(q) \leftarrow \text{CANDIDATEPOOL}(q, m_1, m_2, K)$      ▷ Union of per-encoder top-$K$ lists
3: $\{s_2(q, d_i)\}_{d_i \in \mathcal{C}(q)} \leftarrow \text{SCORE}_{m_2}(q, \mathcal{C}(q))$      ▷ Fixed guidance scores
4: $p_2(d_i \mid e_2^q) \leftarrow \text{softmax}(s_2(q, d_i))$ for $d_i \in \mathcal{C}(q)$      ▷ Normalize $m_2$'s scores over $\mathcal{C}(q)$
5: **for** $t = 0$ **to** $T - 1$ **do**
6:      $p_1(d_i \mid z^{(t)}) \leftarrow \text{softmax}(s_1(z^{(t)}, d_i))$ for $d_i \in \mathcal{C}(q)$      ▷ Primary distribution on $\mathcal{C}(q)$
7:      $p_{\text{avg}}(d_i \mid z^{(t)}) \leftarrow \frac{1}{2}(p_1(d_i \mid z^{(t)}) + p_2(d_i \mid e_2^q))$      ▷ Consensus (average) distribution
8:      $\mathcal{L}_{\text{KL}} \leftarrow \text{KL}(p_{\text{avg}}(d_i \mid z^{(t)}) \| p_1(d_i \mid z^{(t)}))$      ▷ Compute the loss
9:      $z^{(t+1)} \leftarrow z^{(t)} - \alpha \nabla_{z^{(t)}} \mathcal{L}_{\text{KL}}$      ▷ Gradient step on the query representation
10: **end for**
11: $s_1^{(T)}(d_i) \leftarrow s_1(d_i \mid z^{(T)})$ for $d_i \in \mathcal{C}(q)$      ▷ Final primary scores after refinement
12: $\mathcal{R}(q) \leftarrow \text{topK}_{d \in \mathcal{C}(q)} s_1^{(T)}(q, d)$      ▷ Return ordered top-$K$ by score
13: **return** $\mathcal{R}(q)$

---

**Primary and complementary roles** We evaluate text-centric and vision-centric retrievers as both primary and complementary retrievers within GQR, reporting results in Table 19. Across model pairs, both role assignments improve over the base retrievers. The alternative role configuration yields larger gains relative to the primary encoder alone, while the default GQR attains the strongest absolute score on ViDoRe 2.

## F    ADDITIONAL RESULTS

Table 7 presents the results on ViDoRe 1. It is noticeable that this benchmark suffers from saturation, with many subset scores reaching 90 or higher (and indeed this was the direct motivation for the release of ViDoRe 2, Macé et al., 2025). Nevertheless, our method does not harm performance, in contrast to other hybrid retrieval methods, as seen in Tables 8 and 9.

**Storage.** Figure 9 in the Appendix shows that the added storage in GQR is modest. As in the latency plot, GQR dominates the strongest base model by a wide margin. The Llama-Nemo index represents each document with 10.6 MB of memory, whereas the Colnomic hybrid, using Linq as

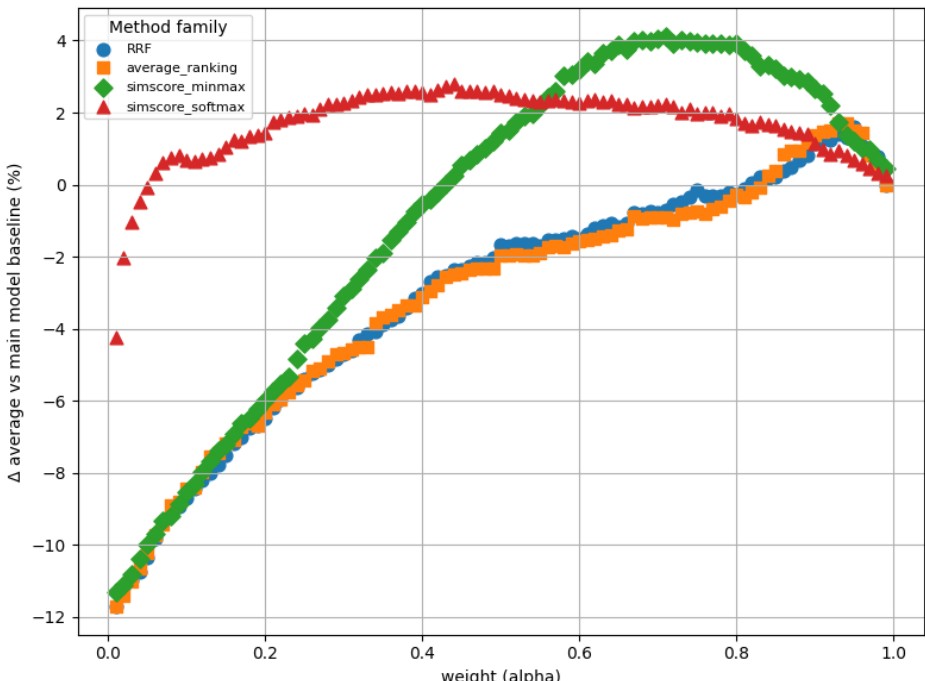

Figure 7: Baseline results on ViDoRe 2 across different values of the weight $\alpha$, averaged over 6 model pairs. It shows that the optimal weight for each method varies. Generally the softmax variant emerges as the most stable, while the min-max variant reaches the best performance if tuned correctly. The values for the zero-shot configuration for each baseline in Table 5 were selected based on this plot.

Table 5: Percentage gain in NDCG@5 of hybrid retrieval over the primary retriever on ViDoRe 3. Each cell reports the average gain and standard deviation over 6 retriever pairs (2 multimodal base retrievers and 3 text retrievers). The *Hyperparameters* column indicates which hyperparameters were selected for zero-shot evaluation.

| Method | Avg | Std | Hyperparameters |
|---|---|---|---|
| Average Ranking (Tuned) | ↑+0.28% | 1.0 | $\alpha = 0.9$ |
| RRF (Tuned) | ↑+0.13% | 0.57 | $\alpha = 0.5$ |
| Score Aggregation (Min-Max, Tuned) | ↑+2.66% | 0.91 | $\alpha = 0.7$ |
| Score Aggregation (Softmax, Tuned) | ↑+1.53% | 1.05 | $\alpha = 0.4$ |
| *Guided Query Refinement (GQR)* | ↑+2.89% | 0.67 | $lr = 10^{-4}$, $T = 50$ |

the refinement model, nearly matches its quality with 0.20 MB per document ($\approx 54\times$ less). Using Jina, GQR surpasses Nemo while requiring 0.37 MB per document ($\approx 28\times$ less).

## G  ANALYZING PER-QUERY DYNAMICS

As we explain in subsection 2.4, the dynamics in GQR differ from a simple weighted average of scores – the distribution changes are constrained by the geometry of the primary model's embedding space, and different documents can shift by different magnitudes. To demonstrate this more directly, in Figure 8 we visualize the changes in document ranks for a given query as a function of the number of GQR steps $t$, and as a function of the weight $\alpha$ of the Score Aggregation (Min-Max) baseline.

Table 6: NDCG@5 for hybrid retrieval on ViDoRe 2. Results are averaged across 9 model pairs.

| Method | Avg | Biomed Lectures | Economics | ESG Human | ESG Full |
|---|---|---|---|---|---|
| Average Ranking | 58.1 | 59.0 | 55.1 | 63.6 | 54.5 |
| RRF | 58.2 | 59.1 | 55.0 | 63.8 | 55.0 |
| Score Aggregation (Min-Max) | 60.2 | 61.3 | 55.8 | 66.7 | 56.8 |
| Score Aggregation (Softmax) | 60.9 | 62.6 | 55.3 | 68.4 | 57.2 |
| Average Ranking - *Tuned* | 59.8 | 61.3 | 55.6 | 66.2 | 56.0 |
| RRF - *Tuned* | 59.9 | 61.4 | 55.5 | 65.3 | 57.3 |
| Score Aggregation (Min-Max) - *Tuned* | 62.1 | 63.9 | **56.5** | 70.0 | 58.0 |
| Score Aggregation (SoftMax) - *Tuned* | 61.6 | 63.4 | 55.4 | 69.5 | 58.2 |
| *Guided Query Refinement (GQR)* | **62.3** | **64.2** | 56.0 | **70.2** | **59.0** |

Table 7: NDCG@5 over ViDoRe 1, by primary and complementary models. Deltas are absolute changes vs. the *No refinement* row within the same base.

| Primary Model | GQR complementary model | Avg val | Avg Δ | ArXivQA val | ArXivQA Δ | DocVQA val | DocVQA Δ | InfoVQA val | InfoVQA Δ | TabFQuAD val | TabFQuAD Δ | TAT DQA val | TAT DQA Δ | ShiftProj val | ShiftProj Δ | SynDocQA AI val | SynDocQA AI Δ | SynDocQA Energy val | SynDocQA Energy Δ | SynDocQA Gov val | SynDocQA Gov Δ | SynDocQA Health val | SynDocQA Health Δ |
|---|---|---|---|---|---|---|---|---|---|---|---|---|---|---|---|---|---|---|---|---|---|---|---|
| **jina (text)** | | 80.9 | 0.0 | 42.9 | 0.0 | 43.7 | 0.0 | 69.2 | 0.0 | 97.6 | 0.0 | 82.4 | 0.0 | 85.6 | 0.0 | 100.0 | 0.0 | 94.4 | 0.0 | 98.9 | 0.0 | 94.4 | 0.0 |
| **Linq-Embed** | | 73.8 | 0.0 | 45.3 | 0.0 | 36.0 | 0.0 | 78.2 | 0.0 | 90.5 | 0.0 | 45.9 | 0.0 | 75.6 | 0.0 | 94.4 | 0.0 | 88.9 | 0.0 | 90.0 | 0.0 | 93.3 | 0.0 |
| **Qwen3** | | 76.4 | 0.0 | 48.2 | 0.0 | 36.9 | 0.0 | 77.2 | 0.0 | 97.2 | 0.0 | 48.7 | 0.0 | 77.8 | 0.0 | 97.8 | 0.0 | 91.1 | 0.0 | 97.8 | 0.0 | 91.1 | 0.0 |
| **Colnomic-7B** | No refinement | 89.8 | 0.0 | **88.6** | 0.0 | 60.2 | 0.0 | **92.4** | 0.0 | 96.7 | 0.0 | 81.2 | 0.0 | **88.7** | 0.0 | **99.6** | 0.0 | **95.9** | 0.0 | **95.1** | 0.0 | **99.2** | 0.0 |
| | **Jina (text)** | 89.7 | ↓-0.1 | 86.9 | ↓-1.7 | **61.3** | ↑+1.1 | **92.4** | 0.0 | 96.7 | 0.0 | 81.2 | 0.0 | **88.7** | 0.0 | **99.6** | 0.0 | **95.9** | 0.0 | **95.1** | 0.0 | **99.2** | 0.0 |
| | **Linq-Embed** | **89.8** | 0.0 | **88.6** | 0.0 | 60.7 | ↑+0.5 | 92.0 | ↓-0.4 | 96.7 | 0.0 | **81.3** | ↑+0.1 | **88.7** | 0.0 | **99.6** | 0.0 | **95.9** | 0.0 | **95.1** | 0.0 | **99.2** | 0.0 |
| | **Qwen3** | **89.8** | 0.0 | 88.4 | ↓-0.2 | 59.9 | ↓-0.3 | **92.4** | 0.0 | **97.2** | ↑+0.5 | 81.2 | 0.0 | **88.7** | 0.0 | **99.6** | 0.0 | **95.9** | 0.0 | **95.1** | 0.0 | **99.2** | 0.0 |
| **Jina (vision)** | No refinement | 89.9 | 0.0 | 88.6 | 0.0 | **62.4** | 0.0 | 92.0 | 0.0 | 96.2 | 0.0 | 78.4 | 0.0 | **91.5** | 0.0 | **99.2** | 0.0 | **96.1** | 0.0 | **96.5** | 0.0 | **98.5** | 0.0 |
| | **Jina (text)** | 89.8 | ↓-0.1 | 87.9 | ↓-0.7 | **62.4** | 0.0 | 92.0 | 0.0 | 96.2 | 0.0 | **78.5** | ↑+0.1 | **91.5** | 0.0 | **99.2** | 0.0 | **96.1** | 0.0 | **96.5** | 0.0 | 97.7 | ↓-0.8 |
| | **Linq-Embed** | 89.8 | ↓-0.1 | 88.8 | ↑+0.2 | 61.0 | ↓-1.4 | **92.1** | ↑+0.1 | 96.2 | 0.0 | 78.4 | 0.0 | **91.5** | 0.0 | **99.2** | 0.0 | **96.1** | 0.0 | **96.5** | 0.0 | **98.5** | 0.0 |
| | **Qwen3** | 89.6 | ↓-0.3 | **88.9** | ↑+0.3 | 59.8 | ↓-2.6 | 92.0 | 0.0 | 96.2 | 0.0 | 78.4 | 0.0 | **91.5** | 0.0 | **99.2** | 0.0 | **96.1** | 0.0 | **96.5** | 0.0 | 97.7 | ↓-0.8 |
| **Llama-Nemo** | No refinement | 91.0 | 0.0 | **88.0** | 0.0 | **66.2** | 0.0 | **94.9** | 0.0 | 96.7 | 0.0 | 81.0 | 0.0 | **89.9** | 0.0 | **100.0** | 0.0 | 96.3 | 0.0 | 97.7 | 0.0 | **99.2** | 0.0 |
| | **Jina (text)** | **91.0** | 0.0 | **88.0** | 0.0 | 65.9 | ↓-0.3 | 94.7 | ↓-0.2 | **97.0** | ↑+0.3 | **81.5** | ↑+0.5 | 89.6 | ↓-0.3 | **100.0** | 0.0 | 96.3 | 0.0 | 97.7 | 0.0 | **99.2** | 0.0 |
| | **Linq-Embed** | **91.0** | 0.0 | **88.0** | 0.0 | 65.8 | ↓-0.4 | 94.5 | ↓-0.4 | 96.9 | ↑+0.2 | 80.9 | ↓-0.1 | 89.7 | ↓-0.2 | **100.0** | 0.0 | **96.7** | ↑+0.4 | **98.1** | ↑+0.4 | **99.2** | 0.0 |
| | **Qwen3** | 90.8 | ↓-0.2 | 87.2 | ↓-0.8 | 65.7 | ↓-0.5 | **94.9** | 0.0 | 96.6 | ↓-0.1 | 80.8 | ↓-0.2 | 89.7 | ↓-0.2 | **100.0** | 0.0 | 95.8 | ↓-0.5 | **98.1** | ↑+0.4 | **99.2** | 0.0 |

As can be seen in the figure, for some queries the dynamics of GQR and score aggregation are relatively similar, whereas for others the effects and dynamics are starkly different. Notably, while score aggregation operates across all candidate documents, the effects of GQR are often focused on a specific subset of documents.

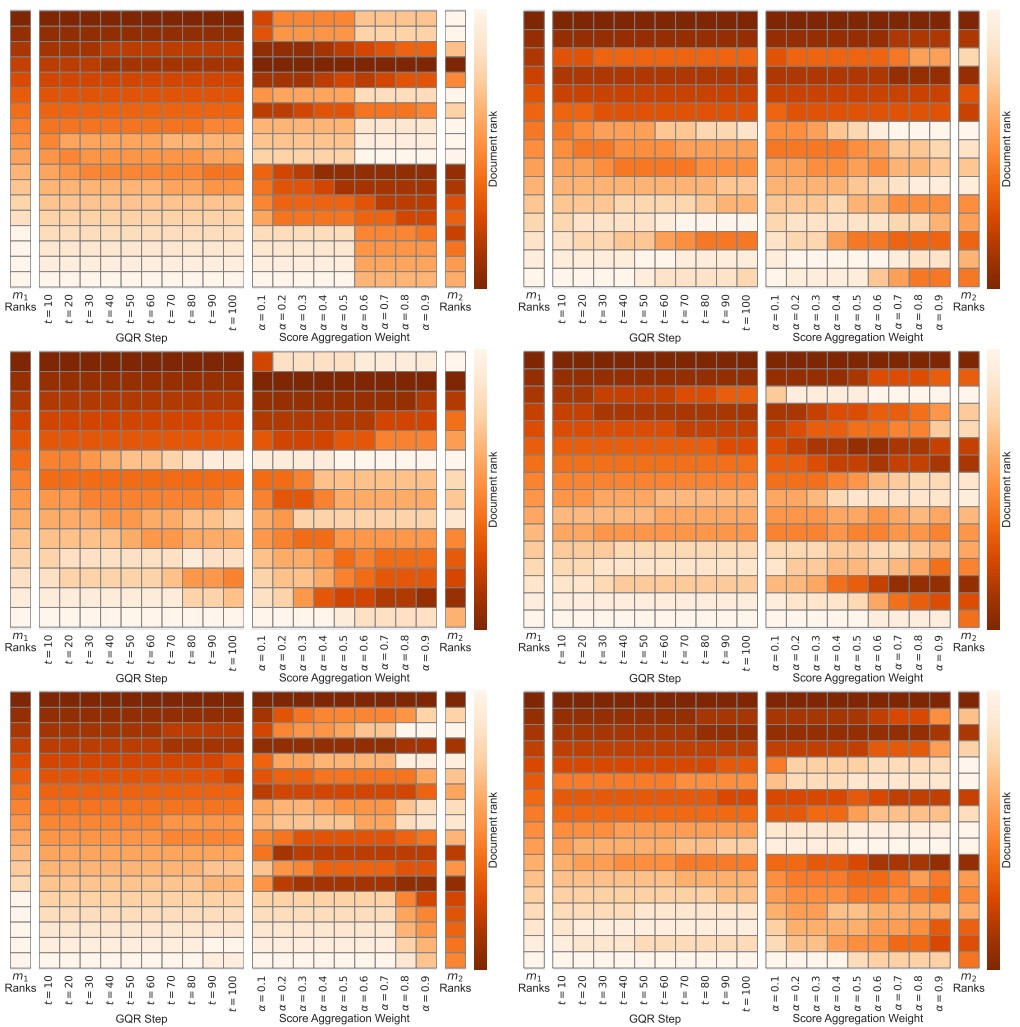

Figure 8: Query-level dynamics of GQR versus score aggregation. The heat maps depict examples of how the document ranks change as a function of either the GQR optimization step $t$ (center left panels), or the weight $\alpha$ of a linear score aggregation baseline (Score Aggregation - Min-Max, center right panels). For both methods we see a gradual progression from the ranks of the primary retriever $m_1$ (left) towards those of the complementary retriever $m_2$ (right). However, the dynamics differ across queries, as well as across documents within a query. Examples are taken from the ESG subset of ViDoRe 2, for the model pair COLNOMIC-EMBED-MULTIMODAL-7B,LINQ-EMBED-MISTRAL (using a GQR learning rate of $10^{-4}$).

Table 8: NDCG@5 for hybrid retrieval on ViDoRe 1. Results are averaged across 9 retriever pairs.

| Method | Avg | arXivQA | DocVQA | InfoVQA | TabFQuAD | TATDQA | ShiftProj | SynthAI | SynthEnergy | SynthGov | SynthHealth |
|---|---|---|---|---|---|---|---|---|---|---|---|
| Average Ranking | 77.8 | 55.6 | 44.0 | 76.2 | 93.4 | 61.1 | 78.5 | 96.0 | 89.8 | 91.0 | 92.7 |
| RRF | 78.0 | 53.8 | 44.1 | 76.5 | 93.5 | 61.7 | 79.0 | 96.1 | 90.3 | 91.4 | 93.3 |
| Score Aggregation (Min-Max) | 84.4 | 74.4 | 53.0 | 85.5 | 95.4 | 69.4 | 83.2 | 97.9 | 94.0 | 95.4 | 95.9 |
| Score Aggregation (Softmax) | 88.6 | 83.3 | 57.7 | 91.0 | 96.5 | 78.3 | 88.2 | **99.6** | 96.0 | **96.6** | 98.6 |
| Average Ranking - *Tuned* | 85.5 | 79.5 | 57.3 | 87.0 | 95.5 | 74.4 | 86.0 | 97.8 | 88.3 | 92.3 | 96.9 |
| RRF - *Tuned* | 84.5 | 76.6 | 55.8 | 85.9 | 95.7 | 73.4 | 84.4 | 97.6 | 88.0 | 92.3 | 95.2 |
| Score Aggregation (Min-Max) - *Tuned* | 88.5 | 88.0 | 62.2 | 92.6 | 96.1 | 79.6 | 88.4 | 97.7 | 88.9 | 93.9 | 98.0 |
| Score Aggregation (SoftMax) - *Tuned* | 89.4 | 87.7 | 62.4 | 92.4 | 96.5 | **80.2** | 86.9 | 99.1 | 95.3 | 96.0 | 97.6 |
| *Guided Query Refinement (GQR)* | **90.1** | **88.1** | **62.5** | **93.0** | **96.6** | 80.2 | **90.0** | **99.6** | **96.1** | 96.5 | **98.8** |

Table 9: Percentage gain, in NDCG@5, of hybrid retrieval over the primary retriever for ViDoRe 1. Each cell depicts average gain over 9 retriever pairs (3 multimodal base retrievers × 3 text retrievers).

| Method | Avg | arXivQA | DocVQA | InfoVQA | TabFQuAD | TATDQA | ShiftProj | SynthAI | SynthEnergy | SynthGov | SynthHealth |
|---|---|---|---|---|---|---|---|---|---|---|---|
| Average Ranking | ↓-14.7% | ↓-37.1% | ↓-29.9% | ↓-18.1% | ↓-3.3% | ↓-23.9% | ↓-12.8% | ↓-3.7% | ↓-6.6% | ↓-5.6% | ↓-6.3% |
| RRF | ↓-14.6% | ↓-39.1% | ↓-29.9% | ↓-17.9% | ↓-3.2% | ↓-23.1% | ↓-12.2% | ↓-3.5% | ↓-6.0% | ↓-5.2% | ↓-5.7% |
| Score Aggregation (Min-Max) | ↓-7.0% | ↓-15.8% | ↓-15.7% | ↓-8.1% | ↓-1.2% | ↓-13.5% | ↓-7.6% | ↓-1.7% | ↓-2.2% | ↓-1.1% | ↓-3.1% |
| Score Aggregation (Softmax) | ↓-2.1% | ↓-5.8% | ↓-8.1% | ↓-2.2% | 0.0 | ↓-2.3% | ↓-2.0% | 0.0 | ↓-0.1% | ↑+0.2% | ↓-0.4% |
| Average Ranking - *Tuned* | ↓-5.5% | ↓-10.1% | ↓-8.9% | ↓-6.6% | ↓-1.0% | ↓-7.2% | ↓-4.5% | ↓-1.8% | ↓-8.1% | ↓-4.3% | ↓-2.1% |
| RRF - *Tuned* | ↓-6.6% | ↓-13.4% | ↓-11.4% | ↓-7.7% | ↓-0.9% | ↓-8.5% | ↓-6.2% | ↓-2.0% | ↓-8.4% | ↓-4.3% | ↓-3.8% |
| Score Aggregation (Min-Max) - *Tuned* | ↓-1.8% | ↓-0.4% | ↓-1.1% | ↓-0.5% | ↓-0.4% | ↓-0.7% | ↓-1.8% | ↓-1.9% | ↓-7.5% | ↓-2.7% | ↓-1.0% |
| Score Aggregation (SoftMax) - *Tuned* | ↓-0.9% | ↓-0.8% | ↓-0.8% | ↓-0.7% | ↓-0.1% | 0.0 | ↓-3.5% | ↓-0.5% | ↓-0.8% | ↓-0.4% | ↓-1.3% |
| *Guided Query Refinement (GQR)* | ↓-0.1% | ↓-0.4% | ↓-0.7% | ↓-0.1% | ↑+0.1% | ↑+0.1% | ↓-0.1% | 0.0 | 0.0 | ↑+0.1% | ↓-0.2% |

Table 10: Performance, latency (ms per query), and memory (MB per document) by primary and complementary models.

| Primary Model | Complementary Model | Performance | Latency | Latency Diff | Memory (MB) |
|---|---|---|---|---|---|
| Colnomic-7b | *No refinement* | 60.25 | 115.98 | | 0.20 |
| Colnomic-7b | Linq-Embed | 62.75 | 181.21 | 65.23 | 0.20 |
| Colnomic-7b | Qwen3 | 60.98 | 196.16 | 80.15 | 0.20 |
| Colnomic-7b | Jina (text) | 63.05 | 350.13 | 194.15 | 0.39 |
| Jina (vision) | *No refinement* | 57.20 | 153.45 | | 0.20 |
| Jina (vision) | Linq-Embed | 61.18 | 213.97 | 60.5 | 0.20 |
| Jina (vision) | Qwen3 | 59.75 | 233.06 | 79.61 | 0.20 |
| Jina (vision) | Jina (text) | 60.68 | 394.64 | 241.19 | 0.39 |
| Llama-Nemo | *No refinement* | 62.98 | 2591.14 | | 11.07 |
| Llama-Nemo | Linq-Embed | 65.15 | 2674.84 | 83.7 | 11.08 |
| Llama-Nemo | Qwen3 | 63.30 | 2712.12 | 120.98 | 11.08 |
| Llama-Nemo | Jina (text) | 64.18 | 2934.61 | 343.47 | 11.27 |

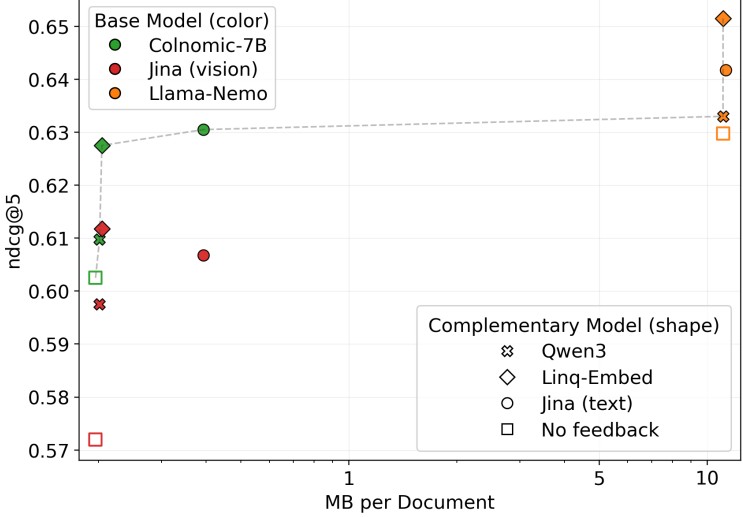

Figure 9: Storage–quality tradeoff. The $x$ axis is memory in MB, on a log scale, and the $y$ axis is the average evaluation score (NDCG@5). Marker color encodes the *primary* retriever; marker shape encodes the GQR *complementary* retriever, with squares indicating the primary retriever alone (without applying GQR).

Table 11: Performance and end-to-end latency of reranking pipelines against GQR. Rows are grouped by reranker candidate size $k$. A dedicated block reports GQR.

| Reranking $k$ | Retriever | Latency | NDCG@5 | Recall@5 |
|---|---|---|---|---|
| No-reranking | Colnomic-7B (multi) | 115.98 | 60.25 | 57.32 |
| | Jina (vision, multi) | 153.45 | 57.20 | 56.17 |
| | Llama Nemo 3B (multi) | 2591.14 | 62.97 | 59.75 |
| **GQR** | Colnomic-7B (multi) | 181.21 | 62.75 | 58.0 |
| | Jina (vision, multi) | 213.97 | 61.0 | 57.6 |
| | Llama Nemo 3B (multi) | 2674.83 | 65.15 | 60.1 |
| 5 | Colnomic-7B (multi) | 1823.03 | 62.12 | 57.32 |
| | Jina (vision, multi) | 1860.714 | 60.70 | 56.17 |
| | Llama Nemo 3B (multi) | 4332.55 | 64.72 | 59.75 |
| 10 | Colnomic-7B (multi) | 3586.809 | 64.37 | 59.92 |
| | Jina (vision, multi) | 3585.946 | 63.27 | 58.8 |
| | Llama Nemo 3B (multi) | 6027.078 | 66.10 | 61.8 |
| 20 | Colnomic-7B (multi) | 7035.953 | 65.07 | 60.27 |
| | Jina (vision, multi) | 7251.81 | 64.10 | 59.6 |
| | Llama Nemo 3B (multi) | 9470.134 | 65.77 | 61.02 |

Table 12: Recall@5 on ViDoRe 2, by primary and complementary models.

| Primary Model | Complementary Model | Avg | Biomed Lectures | Economics | ESG Human | ESG Full |
|---|---|---|---|---|---|---|
| Jina-Embeddings (Text) | | 50.2 | 50.7 | 26.3 | 68.1 | 55.5 |
| Linq-Embed | | 50.1 | 60.3 | 25.7 | 62.7 | 51.8 |
| Qwen3-Embedding | | 44.6 | 57.5 | 25.5 | 53.6 | 41.9 |
| Colnomic-Embed | | 57.3 | 66.9 | 30.9 | 74.2 | 57.3 |
| | Jina-Embeddings (Text) | 58.7 | 67.1 | 30.1 | 74.9 | 62.7 |
| | Linq-Embed | 58.0 | 68.3 | 29.7 | 72.6 | 61.5 |
| | Qwen3-Embedding | 58.7 | 65.1 | 30.8 | 77.3 | 61.4 |
| Jina-Embeddings | | 56.2 | 64.2 | 29.6 | 71.8 | 59.1 |
| | Jina-Embeddings (Text) | 57.5 | 64.2 | 27.7 | 75.9 | 62.1 |
| | Linq-Embed | 57.6 | 66.9 | 29.4 | 72.3 | 61.7 |
| | Qwen3-Embedding | 56.9 | 66.2 | 29.4 | 71.7 | 60.2 |
| Llama-Nemoretriever | | 59.8 | 66.5 | 30.7 | 80.1 | 61.7 |
| | Jina-Embeddings (Text) | 60.2 | 65.9 | 30.5 | 80.1 | 64.1 |
| | Linq-Embed | 60.1 | 68.4 | 28.4 | 79.5 | 64.1 |
| | Qwen3-Embedding | 59.9 | 67.1 | 30.4 | 79.3 | 62.7 |

Table 13: Recall@5 for hybrid retrieval on ViDoRe 2. Results are averaged across 9 retriever pairs.

| Method | Avg | Biomed Lectures | Economics | ESG Human | ESG Full |
|---|---|---|---|---|---|
| Average Ranking | 54.3 | 61.4 | 29.1 | 69.4 | 57.3 |
| RRF | 54.6 | 61.5 | 29.1 | 69.7 | 58.0 |
| Score Aggregation (Min-Max) | 56.7 | 64.7 | 29.7 | 72.3 | 59.9 |
| Score Aggregation (Softmax) | 57.3 | 65.3 | 29.9 | 73.7 | 60.4 |
| Average Ranking - *Tuned* | 56.4 | 64.9 | 29.5 | 71.5 | 59.7 |
| RRF - *Tuned* | 56.3 | 64.6 | 29.5 | 70.5 | 60.7 |
| Score Aggregation (Min-Max) - *Tuned* | 58.5 | 66.9 | 30.3 | 75.5 | 61.4 |
| Score Aggregation (SoftMax) - *Tuned* | 57.9 | 65.8 | 29.6 | 74.5 | 61.6 |
| *Guided Query Refinement (GQR)* | 58.6 | 66.6 | 29.6 | 76.0 | 62.3 |

Table 14: Recall@5 on ViDoRe 1, by primary and complementary models.

| Model | Complementary Model | Avg | arXivQA | DocVQA | InfoVQA | TabFQuAD | TATDQA | ShiftProj | SynthAI | SynthEnergy | SynthGov | SynthHealth |
|---|---|---|---|---|---|---|---|---|---|---|---|---|
| Jina-Embeddings (Text) | | 42.9 | 80.9 | 43.7 | 69.2 | 85.6 | 100.0 | 94.4 | 98.9 | 94.4 | 97.6 | 82.4 |
| Linq-Embed | | 45.3 | 73.8 | 36.0 | 78.2 | 75.6 | 94.4 | 88.9 | 90.0 | 93.3 | 90.5 | 45.9 |
| Qwen3-Embedding | | 48.2 | 76.4 | 36.9 | 77.2 | 77.8 | 97.8 | 91.1 | 97.8 | 91.1 | 97.2 | 48.7 |
| Colnomic-Embed | | 93.1 | 93.7 | 66.2 | 95.5 | 95.6 | 100.0 | 97.8 | 100.0 | 100.0 | 98.8 | 89.5 |
| | Jina-Embeddings (Text) | 90.0 | 93.4 | 67.2 | 95.5 | 95.6 | 100.0 | 97.8 | 100.0 | 100.0 | 98.8 | 89.4 |
| | Linq-Embed | 93.1 | 93.7 | 66.5 | 95.3 | 95.6 | 100.0 | 97.8 | 100.0 | 100.0 | 98.8 | 89.7 |
| | Qwen3-Embedding | 93.1 | 93.8 | 66.3 | 96.0 | 95.6 | 100.0 | 97.8 | 100.0 | 100.0 | 99.2 | 89.6 |
| Jina-Embeddings | | 92.2 | 94.2 | 71.0 | 95.4 | 97.8 | 100.0 | 97.8 | 100.0 | 100.0 | 98.8 | 88.5 |
| | Jina-Embeddings (Text) | 90.9 | 94.0 | 71.0 | 95.4 | 97.8 | 100.0 | 97.8 | 100.0 | 100.0 | 98.8 | 88.5 |
| | Linq-Embed | 92.0 | 94.0 | 69.6 | 95.5 | 97.8 | 100.0 | 97.8 | 100.0 | 100.0 | 98.8 | 88.7 |
| | Qwen3-Embedding | 92.2 | 93.9 | 68.7 | 95.4 | 97.8 | 100.0 | 97.8 | 100.0 | 100.0 | 98.8 | 88.5 |
| Llama-Nemoretriever | | 92.4 | 94.7 | 73.0 | 98.0 | 98.9 | 100.0 | 96.7 | 100.0 | 100.0 | 99.6 | 88.8 |
| | Jina-Embeddings (Text) | 92.4 | 94.6 | 72.7 | 98.0 | 97.8 | 100.0 | 96.7 | 100.0 | 100.0 | 99.6 | 89.0 |
| | Linq-Embed | 92.4 | 94.8 | 72.4 | 97.6 | 98.9 | 100.0 | 97.8 | 100.0 | 100.0 | 99.6 | 88.9 |
| | Qwen3-Embedding | 91.6 | 94.5 | 71.5 | 98.2 | 98.9 | 100.0 | 96.7 | 100.0 | 100.0 | 98.8 | 88.8 |

Table 15: Recall@5 for hybrid retrieval on ViDoRe 1. Results are averaged across 9 retriever pairs.

| Method | Avg | arXivQA | DocVQA | InfoVQA | TabFQuAD | TATDQA | ShiftProj | SynthAI | SynthEnergy | SynthGov | SynthHealth |
|---|---|---|---|---|---|---|---|---|---|---|---|
| Average Ranking | 84.1 | 63.9 | 51.8 | 81.5 | 97.1 | 71.6 | 87.8 | 99.3 | 93.7 | 98.0 | 96.7 |
| RRF | 83.8 | 59.5 | 51.3 | 81.8 | 97.2 | 72.4 | 88.4 | 99.0 | 93.7 | 98.3 | 96.7 |
| Score Aggregation (Min-Max) | 91.5 | 88.4 | 63.5 | 93.1 | 98.6 | 83.0 | 92.3 | 100.0 | 97.6 | 99.9 | 98.4 |
| Score Aggregation (Softmax) | 92.5 | 87.4 | 63.5 | 94.5 | 99.1 | 87.0 | 96.1 | 100.0 | 97.3 | 100.0 | 100.0 |
| Average Ranking - *Tuned* | 92.9 | 92.2 | 68.7 | 95.6 | 98.8 | 87.6 | 95.2 | 99.9 | 93.3 | 98.2 | 100.0 |
| RRF - *Tuned* | 91.5 | 90.9 | 66.2 | 93.5 | 98.8 | 85.0 | 92.4 | 99.5 | 92.8 | 98.2 | 97.2 |
| Score Aggregation (Min-Max) - *Tuned* | 93.5 | 92.3 | 69.3 | 96.3 | 99.0 | 88.7 | 96.6 | 99.5 | 93.8 | 99.1 | 100.0 |
| Score Aggregation (SoftMax) - *Tuned* | 93.5 | 91.4 | 69.4 | 95.8 | 99.0 | 88.9 | 94.7 | 99.9 | 97.1 | 99.6 | 98.8 |
| *Guided Query Refinement (GQR)* | 94.1 | 92.0 | 69.5 | 96.3 | 99.0 | 89.0 | 97.3 | 100.0 | 97.6 | 100.0 | 100.0 |

Table 16: Effect of extra index search on GQR NDCG@5 performance, over ViDoRe 2.

| Model | Complementary Model | Variant | Avg | Biomed Lectures | Economics | ESG Human | ESG Full |
|---|---|---|---|---|---|---|---|
| Colnomic-Embed | Jina-Embeddings | GQR | 63.0 | 64.7 | 57.0 | 70.3 | 60.2 |
| | | GQR + *Search* | 63.1 | 64.7 | 57.1 | 70.3 | 60.2 |
| | Linq-Embed | GQR | 62.8 | 65.4 | 56.7 | 67.7 | 61.2 |
| | | GQR + *Search* | 62.7 | 65.4 | 56.5 | 67.9 | 61.0 |
| | Qwen3-Embedding | GQR | 61.0 | 61.9 | 54.3 | 70.2 | 57.5 |
| | | GQR + *Search* | 61.0 | 61.7 | 54.3 | 70.6 | 57.5 |
| Jina-Embeddings | Jina-Embeddings | GQR | 60.7 | 61.7 | 55.3 | 66.9 | 58.8 |
| | | GQR + *Search* | 60.7 | 61.7 | 55.2 | 66.9 | 58.9 |
| | Linq-Embed | GQR | 61.2 | 64.7 | 57.2 | 65.7 | 57.1 |
| | | GQR + *Search* | 61.0 | 64.7 | 57.2 | 65.0 | 57.2 |
| | Qwen3-Embedding | GQR | 59.8 | 63.2 | 53.6 | 67.8 | 54.4 |
| | | GQR + *Search* | 59.8 | 63.2 | 53.6 | 67.8 | 54.4 |
| Llama-Nemoretriever | Jina-Embeddings | GQR | 64.2 | 64.5 | 57.6 | 74.2 | 60.4 |
| | | GQR + *Search* | 64.1 | 64.4 | 57.6 | 74.2 | 60.4 |
| | Linq-Embed | GQR | 65.1 | 66.4 | 56.8 | 74.6 | 62.8 |
| | | GQR + *Search* | 65.3 | 66.5 | 57.2 | 74.6 | 62.8 |
| | Qwen3-Embedding | GQR | 63.3 | 65.0 | 55.4 | 74.1 | 58.7 |
| | | GQR + *Search* | 63.3 | 64.8 | 55.4 | 74.1 | 58.7 |

Table 17: Effect of candidate pool on GQR NDCG@5 performance, over ViDoRe 2.

| Model | Complementary Model | Variant | Avg | Biomed Lectures | Economics | ESG Human | ESG Full |
|---|---|---|---|---|---|---|---|
| Colnomic-Embed | Jina-Embeddings | GQR | 63.0 | 64.7 | 57.0 | 70.3 | 60.2 |
| | | GQR (Top-$K$ only) | 62.9 | 64.4 | 56.3 | 71.2 | 59.6 |
| | Linq-Embed | GQR | 62.8 | 65.4 | 56.7 | 67.7 | 61.2 |
| | | GQR (Top-$K$ only) | 62.4 | 63.5 | 57.3 | 68.6 | 60.1 |
| | Qwen3-Embedding | GQR | 61.0 | 61.9 | 54.3 | 70.2 | 57.5 |
| | | GQR (Top-$K$ only) | 61.5 | 63.4 | 54.3 | 71.6 | 56.7 |
| Jina-Embeddings | Jina-Embeddings | GQR | 60.7 | 61.7 | 55.3 | 66.9 | 58.8 |
| | | GQR (Top-$K$ only) | 59.7 | 61.7 | 55.1 | 65.0 | 56.8 |
| | Linq-Embed | GQR | 61.2 | 64.7 | 57.2 | 65.7 | 57.1 |
| | | GQR (Top-$K$ only) | 61.0 | 64.7 | 56.9 | 66.2 | 56.2 |
| | Qwen3-Embedding | GQR | 59.8 | 63.2 | 53.6 | 67.8 | 54.4 |
| | | GQR (Top-$K$ only) | 59.0 | 63.5 | 53.3 | 65.0 | 54.0 |
| Llama-Nemoretriever | Jina-Embeddings | GQR | 64.2 | 64.5 | 57.6 | 74.2 | 60.4 |
| | | GQR (Top-$K$ only) | 64.2 | 64.4 | 57.9 | 74.1 | 60.3 |
| | Linq-Embed | GQR | 65.1 | 66.4 | 56.8 | 74.6 | 62.8 |
| | | GQR (Top-$K$ only) | 64.7 | 65.4 | 57.7 | 74.8 | 60.8 |
| | Qwen3-Embedding | GQR | 63.3 | 65.0 | 55.4 | 74.1 | 58.7 |
| | | GQR (Top-$K$ only) | 63.6 | 65.2 | 56.9 | 74.1 | 58.2 |

Table 18: Effect of loss function on GQR NDCG@5 performance, over ViDoRe 2.

| Model | Complementary Model | Loss Variant | Avg | Biomed Lectures | Economics | ESG Human | ESG Full |
|---|---|---|---|---|---|---|---|
| Colnomic-Embed | Jina-Embeddings | Jensen–Shannon | 62.7 | 64.6 | 56.6 | 69.5 | 60.2 |
| | | Kullback–Leibler (Consensus) | 63.0 | 64.7 | 57.0 | 70.3 | 60.2 |
| | | Kullback–Leibler (Target) | 62.1 | 64.6 | 53.3 | 70.6 | 60.1 |
| | Linq-Embed | Jensen–Shannon | 63.3 | 65.3 | 54.9 | 71.3 | 61.9 |
| | | Kullback–Leibler (Consensus) | 62.8 | 65.4 | 56.7 | 67.7 | 61.2 |
| | | Kullback–Leibler (Target) | 63.8 | 64.9 | 57.3 | 71.3 | 61.7 |
| | Qwen3-Embedding | Jensen–Shannon | 61.4 | 63.6 | 54.3 | 70.5 | 57.1 |
| | | Kullback–Leibler (Consensus) | 61.0 | 61.9 | 54.3 | 70.2 | 57.5 |
| | | Kullback–Leibler (Target) | 61.3 | 64.2 | 54.3 | 70.1 | 56.7 |
| Jina-Embeddings | Jina-Embeddings | Jensen–Shannon | 60.3 | 61.7 | 56.1 | 67.1 | 56.5 |
| | | Kullback–Leibler (Consensus) | 60.7 | 61.7 | 55.3 | 66.9 | 58.8 |
| | | Kullback–Leibler (Target) | 60.9 | 61.7 | 55.3 | 68.5 | 57.9 |
| | Linq-Embed | Jensen–Shannon | 62.5 | 63.7 | 58.7 | 69.8 | 57.8 |
| | | Kullback–Leibler (Consensus) | 61.2 | 64.7 | 57.2 | 65.7 | 57.1 |
| | | Kullback–Leibler (Target) | 61.5 | 64.7 | 55.5 | 68.9 | 57.1 |
| | Qwen3-Embedding | Jensen–Shannon | 59.0 | 62.9 | 53.5 | 64.7 | 54.7 |
| | | Kullback–Leibler (Consensus) | 59.8 | 63.2 | 53.6 | 67.8 | 54.4 |
| | | Kullback–Leibler (Target) | 59.5 | 63.5 | 51.9 | 67.4 | 55.2 |
| Llama-Nemoretriever | Jina-Embeddings | Jensen–Shannon | 64.0 | 64.1 | 57.3 | 74.3 | 60.4 |
| | | Kullback–Leibler (Consensus) | 64.2 | 64.5 | 57.6 | 74.2 | 60.4 |
| | | Kullback–Leibler (Target) | 64.2 | 64.6 | 57.2 | 74.3 | 60.6 |
| | Linq-Embed | Jensen–Shannon | 65.0 | 66.0 | 56.3 | 74.6 | 63.1 |
| | | Kullback–Leibler (Consensus) | 65.1 | 66.4 | 56.8 | 74.6 | 62.8 |
| | | Kullback–Leibler (Target) | 64.7 | 66.2 | 56.7 | 74.3 | 61.5 |
| | Qwen3-Embedding | Jensen–Shannon | 63.4 | 65.1 | 55.7 | 74.1 | 58.6 |
| | | Kullback–Leibler (Consensus) | 63.3 | 65.0 | 55.4 | 74.1 | 58.7 |
| | | Kullback–Leibler (Target) | 63.3 | 65.1 | 55.4 | 74.1 | 58.7 |

Table 19: Swapping primary and complementary roles in GQR across model pairs. The first two columns specify the role of each encoder. For each setting we report the absolute score on ViDoRe 2 and the absolute gain relative to the primary encoder alone.

| Primary model | Complementary model | NDCG@5 | Gain |
|---|---|---|---|
| Colnomic-7B | Jina (text) | 63.05 | 2.8 |
| Jina (text) | Colnomic-7B | 62.22 | 8.82 |
| Colnomic-7B | Linq-Embed | 62.75 | 2.5 |
| Linq-Embed | Colnomic-7B | 61.3 | 6 |
| Colnomic-7B | Qwen 3 | 60.97 | 0.7 |
| Qwen3 | Colnomic-7B | 54.4 | 7.6 |
| Jina (vision) | Jina (text) | 60.67 | 3.5 |
| Jina (text) | Jina (vision) | 59.25 | 5.85 |
| Jina (vision) | Linq-Embed | 61.17 | 4.0 |
| Linq-Embed | Jina (vision) | 61.37 | 6.07 |
| Jina (vision) | Qwen3 | 61.17 | 2.6 |
| Qwen3 | Jina (vision) | 52.05 | 5.25 |
| Llama-Nemo | Jina (text) | 64.17 | 1.2 |
| Jina (text) | Llama-Nemo | 59.3 | 5.9 |
| Llama-Nemo | Linq-Embed | 65.15 | 2.2 |
| Linq-Embed | Llama-Nemo | 60.27 | 4.97 |
| Llama-Nemo | Qwen3 | 63.3 | 0.3 |
| Qwen3 | Llama-Nemo | 52.8 | 6 |

Table 20: Tuned values for the GQR step size parameter $\alpha$ (learning rate) over ViDoRe 2.

| Primary model | Complementary model | Biomedical | Economics | ESG | ESG Human |
|---|---|---|---|---|---|
| | | | Selected $\alpha$ value | | |
| Colnomic-Embed-Multimodal-7B | Jina-Embeddings-V4 - Text | $5 \times 10^{-4}$ | $5 \times 10^{-3}$ | $1 \times 10^{-3}$ | $5 \times 10^{-4}$ |
| | Linq-Embed-Mistral | $5 \times 10^{-4}$ | $5 \times 10^{-3}$ | $1 \times 10^{-3}$ | $5 \times 10^{-3}$ |
| | Qwen3-Embedding-4B | $1 \times 10^{-3}$ | $1 \times 10^{-3}$ | $1 \times 10^{-5}$ | $1 \times 10^{-4}$ |
| Jina-Embeddings-V4 | Jina-Embeddings-V4 - Text | $1 \times 10^{-5}$ | $1 \times 10^{-3}$ | $1 \times 10^{-3}$ | $5 \times 10^{-4}$ |
| | Linq-Embed-Mistral | $5 \times 10^{-4}$ | $1 \times 10^{-3}$ | $1 \times 10^{-4}$ | $1 \times 10^{-3}$ |
| | Qwen3-Embedding-4B | $5 \times 10^{-4}$ | $1 \times 10^{-5}$ | $5 \times 10^{-5}$ | $1 \times 10^{-3}$ |
| Llama-Nemoretriever-Colembed-3B-V1 | Jina-Embeddings-V4 - Text | $5 \times 10^{-5}$ | $1 \times 10^{-5}$ | $1 \times 10^{-5}$ | $1 \times 10^{-5}$ |
| | Linq-Embed-Mistral | $1 \times 10^{-4}$ | $5 \times 10^{-5}$ | $1 \times 10^{-4}$ | $1 \times 10^{-5}$ |
| | Qwen3-Embedding-4B | $1 \times 10^{-4}$ | $5 \times 10^{-5}$ | $1 \times 10^{-5}$ | $1 \times 10^{-5}$ |

Table 21: Tuned values for the number of GQR optimization steps parameter $T$ over ViDoRe 2.

| Primary model | Complementary model | Biomedical | Economics | ESG | ESG Human |
|---|---|---|---|---|---|
| | | | Selected $T$ value | | |
| Colnomic-Embed-Multimodal-7B | Jina-Embeddings-V4 - Text | 10 | 25 | 25 | 25 |
| | Linq-Embed-Mistral | 25 | 25 | 50 | 10 |
| | Qwen3-Embedding-4B | 25 | 10 | 10 | 50 |
| Jina-Embeddings-V4 | Jina-Embeddings-V4 - Text | 10 | 50 | 25 | 25 |
| | Linq-Embed-Mistral | 25 | 25 | 50 | 25 |
| | Qwen3-Embedding-4B | 10 | 50 | 50 | 10 |
| Llama-Nemoretriever-Colembed-3B-V1 | Jina-Embeddings-V4 - Text | 25 | 50 | 50 | 10 |
| | Linq-Embed-Mistral | 10 | 50 | 10 | 10 |
| | Qwen3-Embedding-4B | 10 | 10 | 50 | 10 |

Table 22: Tuned values for the weight parameter $\alpha$ of the Average Ranking baseline over ViDoRe 2.

| Primary model | Complementary model | Biomedical | Economics | ESG | ESG Human |
|---|---|---|---|---|---|
| | | | Selected $\alpha$ value | | |
| Colnomic-Embed-Multimodal-7B | Jina-Embeddings-V4 - Text | 0.8 | 0.7 | 0.6 | 0.8 |
| | Linq-Embed-Mistral | 0.4 | 0.5 | 0.7 | 0.8 |
| | Qwen3-Embedding-4B | 0.9 | 0.9 | 0.9 | 0.8 |
| Jina-Embeddings-V4 | Jina-Embeddings-V4 - Text | 0.9 | 0.5 | 0.5 | 0.8 |
| | Linq-Embed-Mistral | 0.7 | 0.3 | 0.9 | 0.2 |
| | Qwen3-Embedding-4B | 0.9 | 0.9 | 0.8 | 0.5 |
| Llama-Nemoretriever-Colembed-3B-V1 | Jina-Embeddings-V4 - Text | 0.8 | 0.9 | 0.9 | 0.9 |
| | Linq-Embed-Mistral | 0.9 | 0.5 | 0.9 | 0.9 |
| | Qwen3-Embedding-4B | 0.9 | 0.9 | 0.8 | 0.9 |

Table 23: Tuned values for the weight parameter $\alpha$ of the RRF baseline over ViDoRe 2.

| Primary model | Complementary model | Biomedical | Economics | ESG | ESG Human |
|---|---|---|---|---|---|
| | | | Selected $\alpha$ value | | |
| Colnomic-Embed-Multimodal-7B | Jina-Embeddings-V4 - Text | 0.9 | 0.7 | 0.6 | 0.9 |
| | Linq-Embed-Mistral | 0.5 | 0.5 | 0.9 | 0.7 |
| | Qwen3-Embedding-4B | 0.9 | 0.9 | 0.9 | 0.6 |
| Jina-Embeddings-V4 | Jina-Embeddings-V4 - Text | 0.9 | 0.5 | 0.5 | 0.8 |
| | Linq-Embed-Mistral | 0.7 | 0.3 | 0.8 | 0.6 |
| | Qwen3-Embedding-4B | 0.9 | 0.9 | 0.8 | 0.8 |
| Llama-Nemoretriever-Colembed-3B-V1 | Jina-Embeddings-V4 - Text | 0.8 | 0.9 | 0.9 | 0.9 |
| | Linq-Embed-Mistral | 0.9 | 0.4 | 0.9 | 0.9 |
| | Qwen3-Embedding-4B | 0.9 | 0.9 | 0.9 | 0.9 |

Table 24: Tuned values for the weight parameter $\alpha$ of the Score Aggregation (Min-Max) baseline over ViDoRe 2.

| Primary model | Complementary model | Selected $\alpha$ value | | | |
| --- | --- | --- | --- | --- | --- |
| | | Biomedical | Economics | ESG | ESG Human |
| Colnomic-Embed-Multimodal-7B | Jina-Embeddings-V4 - Text | 0.8 | 0.4 | 0.7 | 0.8 |
| | Linq-Embed-Mistral | 0.6 | 0.5 | 0.6 | 0.8 |
| | Qwen3-Embedding-4B | 0.7 | 0.8 | 0.9 | 0.7 |
| Jina-Embeddings-V4 | Jina-Embeddings-V4 - Text | 0.8 | 0.7 | 0.6 | 0.8 |
| | Linq-Embed-Mistral | 0.7 | 0.5 | 0.8 | 0.6 |
| | Qwen3-Embedding-4B | 0.8 | 0.9 | 0.8 | 0.6 |
| Llama-Nemoretriever-Colembed-3B-V1 | Jina-Embeddings-V4 - Text | 0.8 | 0.7 | 0.8 | 0.9 |
| | Linq-Embed-Mistral | 0.7 | 0.8 | 0.7 | 0.9 |
| | Qwen3-Embedding-4B | 0.8 | 0.9 | 0.9 | 0.9 |

Table 25: Tuned values for the weight parameter $\alpha$ of the Score Aggregation (Softmax) baseline over ViDoRe 2.

| Primary model | Complementary model | Selected $\alpha$ value | | | |
| --- | --- | --- | --- | --- | --- |
| | | Biomedical | Economics | ESG | ESG Human |
| Colnomic-Embed-Multimodal-7B | Jina-Embeddings-V4 - Text | 0.6 | 0.1 | 0.5 | 0.7 |
| | Linq-Embed-Mistral | 0.1 | 0.2 | 0.1 | 0.1 |
| | Qwen3-Embedding-4B | 0.1 | 0.2 | 0.4 | 0.1 |
| Jina-Embeddings-V4 | Jina-Embeddings-V4 - Text | 0.9 | 0.5 | 0.6 | 0.6 |
| | Linq-Embed-Mistral | 0.2 | 0.1 | 0.2 | 0.1 |
| | Qwen3-Embedding-4B | 0.6 | 0.5 | 0.9 | 0.1 |
| Llama-Nemoretriever-Colembed-3B-V1 | Jina-Embeddings-V4 - Text | 0.8 | 0.8 | 0.9 | 0.9 |
| | Linq-Embed-Mistral | 0.1 | 0.1 | 0.1 | 0.9 |
| | Qwen3-Embedding-4B | 0.2 | 0.6 | 0.8 | 0.9 |

