# OpenReview forum: "Guided Query Refinement: Multimodal Hybrid Retrieval with Test-Time Optimization"
_ICLR.cc/2026/Conference — ICLR 2026 Poster_

### Official Review · Reviewer_EqPn · 2025-10-28

**Soundness:** 2
**Presentation:** 3
**Contribution:** 2
**Rating:** 4
**Confidence:** 4

**Summary:**

Guided Query Refinement is a test-time retrieval ensembling method that iteratively updates the primary retriever’s query representation using gradient descent to minimize the KL divergence of the (query, top k document) probability distribution to the secondary retriever's distribution. On the ViDoRe 2 benchmark,  it outperforms other ensembling methods, as well as the primary retriever when used on its own.

**Strengths:**

This paper provides an interesting and novel framework to refine query representations at test time through gradient descent on the representation itself, in order to leverage information from a complementary retriever. The method used enables optimizing multi and single vector models, and yields strong results on Vidore V2.

The paper is mostly well written and relatively easy to understand.

**Weaknesses:**

**Objective function simplification**: The objective function that is minimized here admits a "trivial" minimum when p1(t) = p2. While this may be hard to reach in practice, the role of the "extra-step" of averaging p1(t) and p2 is unclear to me: since p_avg(t) depends on p1(t), it just seems that by minimizing KL(p_avg(t) || p1(t)), we are in practice (mostly) minimimizing KL(p2||p1(t)), and thus nudging p1(t) to match the p2 distribution. The ablations that are run l421 mostly confirm this is the case.  Under this light, it is not clear what the additional complexity of the "averaging step" brings.

**Objective function rationale**: In all cases, it seems that the effect of GQR is to modify the query representation(s) to more closely match the probability distribution yielded by a weaker secondary retriever. I personally find hard to gain intuitions on why this would help in practice - and how this differs from some sort of tuned probability distribution averaging method.

I would find interesting to understand how scores evolve through training steps: intuitively, at t=0, we match the primary model nDCG, then (according to your results), the ranking is slightly improved, and I would imagine that eventually, as more steps are made and p1(t) too closely matches p2, performance drops and starts matching those of the secondary models ? Understanding these dynamics could be key here in explaining why this method works. It would also be interesting to have some plots of the score distributions before/after GQR to better understand what is going on. Finally, a measure of how much p1(0) and p2 impact the final distribution (ratio of KL divs for instance) could be given over the different data subsets on which hyperparam optimization is run to gain a sense of how data dependant GQR is.

**Hyperparameter sensitivity**:  From my understanding, the dynamic described above would indicate that there is a bounded ranges of optimization step that can be made before collapse, and thus that GQR is sensitive to hyperparameter choice. It also seems you independently optimize parameters for each data subset in Vidore 2. Would it be possible to show results where hyperparameters are not optimized on the same data distribution (ie. optimize on Vidore train set, test on the full Vidore v2 with the same parameters across each split ?).

*I believe it is key to add the final hyperparameter values (for the method as well as the baselines) used in the appendix. Not having them makes assessing the work much more difficult.*

**Missing files**: The anonymous4science link works (and is appreciated) but most files are missing. I can only see the Readme and the latency benchmarking codefiles, not the rest. I also don't see the chosen hyperparams.

**Clarity**: In this paper, the query embeddings that are optimized are multi-vector embeddings (in most cases). While the query refinement technique applies directly here, this could be more clearly explicited - I feel this "form agnosticity" is one of the strenghts of this method.

**Overall**: I am aware I suggest many things here, the overall point is that the rationale behind "why" GQR works remains unclear, and understanding the method's sensitivity to hyperparameters would be key to inform practical uses of this method beyond benchmark maximisation.

**Questions:**

1.  (See *Weakness Objective function simplification*) Is there any reason to justify the additional complexity of the "averaging step" in the objective function that seems redundant ?

2. You state that hyperparameters are chosen on a dev split of 10% of the data. To my knowledge, no dev split exists for Vidore V2. Do you thus split the test split and report results on 90% of the remaining docs ? If this is the case, is there any overlap between docs in the "test" and your current "dev" split here ?

3. How do you explain GQR does not improve results on ViDoRe 1 ?

4. Under the "nudging towards p2" hypothesis, GQR would be ~roughly~ equivalent to a "tuned" normalized score averaging technique in which the tuning is done through gradient descent. In your results, the best baselines in practice are also "tuned" normalized score averaging, but the alpha parameter that can be optimized is discrete (steps of 0.1). Could the performance gap be closed if alpha becomes a learnable parameter here (or at least is optimized at a more granular level) ? Could you at least report alpha per subset ?

**Suggestions**:
Seems figures are big enough to integrate text from the legends directly in the plot. This may make it a bit easier to understand the plots in one glance. This is very minor of course.
Add x-axis label in figure 3.

---

> ### Author Response · Authors · 2025-11-22
> **Part 1**
>
> We thank the reviewer for their detailed comments and questions about our work!
> We appreciate the depth of thought and effort reflected in the review.
> We address comments and questions below, and would be happy to engage further during the discussion phase!
>
>
> > Objective function simplification: The objective function that is minimized here admits a "trivial" minimum when p1(t) = p2. While this may be hard to reach in practice, the role of the "extra-step" of averaging p1(t) and p2 is unclear to me: since p_avg(t) depends on p1(t), it just seems that by minimizing KL(p_avg(t) || p1(t)), we are in practice (mostly) minimimizing KL(p2||p1(t)), and thus nudging p1(t) to match the p2 distribution. The ablations that are run l421 mostly confirm this is the case. Under this light, it is not clear what the additional complexity of the "averaging step" brings.
>
> We added Section 2.4 to clarify the rationale behind our objective. In standard distillation setups, KL is minimized directly between the teacher and student, yet our setup is different since the complementary retriever providing the signal is on par or weaker. Given two equally strong models, a natural choice is to define the teacher distribution as the average of the two (ensemble). It is true that when t->infinity, p1(t) = p2 and by minimizing KL(p_avg(t) || p1(t)) GQR also minimizes KL(p2||p1(t)). However, during the optimization trajectory, the two objectives provide meaningfully different gradients, and at a fixed T value the query representation might vary significantly between the two. We chose  KL(p_avg(t) || p1(t)) as the objective behind GQR to encourage reduced variance and a more stable optimization process.  While we found in the ablations that GQR **with hyperparameter tuning** can achieve good results even with minimizing KL(p2||p1(t)), we have since followed up with a more thorough hyperparameter investigation and found that KL(p_avg(t) || p1(t)) is a generally more stable objective across hyperparameters. We will follow up with including these additional empirical results in the paper.
>
>
> > Objective function rationale: In all cases, it seems that the effect of GQR is to modify the query representation(s) to more closely match the probability distribution yielded by a weaker secondary retriever. I personally find hard to gain intuitions on why this would help in practice - and how this differs from some sort of tuned probability distribution averaging method.
>
> We added Section 2.4 to also attempt and clarify the motivation for query refinement. The score updates in GQR are the result of query-document similarity with the updated query, and since the documents are fixed in this space, the scores are constrained to the original geometry. This means that the feedback of the complementary retriever is incorporated more softly and with alignment to the primary retriever. As an intuitive example, one can consider two score distributions - p_1=[0.4,0.4,0.2], p_2=[0.33,0.33,0.33]. In score-level fusion, p_1 is nudged towards p_2 in equal magnitudes for each document, so the result is always a simple weighted average (affine combination). In GQR, it is possible that the geometry of the space will nudge each document score in different magnitudes and in a non-linear way, so after GQR the scores will be p_gqr=[0.34,38,28] (e.g. the first document is easily nudged toward p_2 while the second one is not easily changed, due to the inductive bias in the primary retriever's space).
>
>
>
> > I would find interesting to understand how scores evolve through training steps: intuitively, at t=0, we match the primary model nDCG, then (according to your results), the ranking is slightly improved, and I would imagine that eventually, as more steps are made and p1(t) too closely matches p2, performance drops and starts matching those of the secondary models ? Understanding these dynamics could be key here in explaining why this method works. It would also be interesting to have some plots of the score distributions before/after GQR to better understand what is going on. Finally, a measure of how much p1(0) and p2 impact the final distribution (ratio of KL divs for instance) could be given over the different data subsets on which hyperparam optimization is run to gain a sense of how data dependant GQR is.
>
> Besides the rationale provided above and in Section 2.4, we are currently working on providing more analysis of GQR. We will share our it as soon as we can.

---

> > ### Author Response · Authors · 2025-11-22
> > **Part 2**
> >
> > > Hyperparameter sensitivity: From my understanding, the dynamic described above would indicate that there is a bounded ranges of optimization step that can be made before collapse, and thus that GQR is sensitive to hyperparameter choice. It also seems you independently optimize parameters for each data subset in Vidore 2. Would it be possible to show results where hyperparameters are not optimized on the same data distribution (ie. optimize on Vidore train set, test on the full Vidore v2 with the same parameters across each split ?).
> >
> > Following these suggestions, and aiming to better understand the sensitivity to parameter choice, **we have added a sub-section in the analysis (§4.1, Figure 4) which investigates the relation between the hyperparameter values T and $\alpha$ and the resulting GQR performance.** This analysis demonstrates that for medium learning rates the effect of the method is quite stable for a range of different values of T.
> > Informed by this analysis, in the added results section “3.3 Results – Zero-Shot Performance”, we run GQR on the newly-released ViDoRe V3 benchmark, **using a single hyperparameter configuration across 6 model pairs and 8 subsets. We find that GQR is highly effective, improving performance for all pairs in a zero-shot setting, and achieving state-of-the-art performance on this unseen benchmark without any parameter tuning on its specific data distribution.**
> >
> > > I believe it is key to add the final hyperparameter values (for the method as well as the baselines) used in the appendix. Not having them makes assessing the work much more difficult.
> >
> > We added the selected hyperparameter values in Tables 19-24. In the hyperparameter investigation (sub-section 4.1), we explicitly provide recommended hyperparameter configurations for zero-shot usage.
> >
> >
> > > The anonymous4science link works (and is appreciated) but most files are missing. I can only see the Readme and the latency benchmarking codefiles, not the rest. I also don't see the chosen hyperparams.
> >
> > We are unsure what happened there, as far as we can tell all the files are there in the link (perhaps there was a temporary issue with the site?). Specifically, the main script is `run_experiments.py` (this script also includes the set of possible hyperparameter options);  the GQR algorithm is there under `query_optimizations.py`, the models under `embedding_configs.py` and so on. Regarding the chosen (tuned) hyperparameters, we have now added those to the paper as requested.
> >
> > > Clarity: In this paper, the query embeddings that are optimized are multi-vector embeddings (in most cases). While the query refinement technique applies directly here, this could be more clearly explicited - I feel this "form agnosticity" is one of the strenghts of this method.
> >
> > We agree. We added this emphasis to line ~97 of the introduction and to the method section in line ~190.
> >
> >
> > > You state that hyperparameters are chosen on a dev split of 10% of the data. To my knowledge, no dev split exists for Vidore V2. Do you thus split the test split and report results on 90% of the remaining docs ? If this is the case, is there any overlap between docs in the "test" and your current "dev" split here ?
> >
> >
> > There is no official dev set for ViDoRe2 and so we split the benchmark into dev and test randomly and report results on the test. We do a queries-based split. 10% of the queries are marked as dev, and the different methods are tuned on top of them. Splitting based on documents introduces additional complexity since the queries in the benchmark have several gold_id documents labeled for each.
> >
> > > How do you explain GQR does not improve results on ViDoRe 1 ?
> >
> >
> > The benchmark is saturated and so we believe that it is too noisy to provide a reliable signal.
> > As stated before, in this version we additionally add results on ViDoRe3 in zero-shot settings, and find significant improvements across all subsets and model pairs, reinforcing this claim. Notably, GQR maintains performance on ViDoRe 1 and doesn't harm it.
> >
> > > Under the "nudging towards p2" hypothesis, GQR would be roughly equivalent to a "tuned" normalized score averaging technique in which the tuning is done through gradient descent. Could the performance gap be closed if alpha becomes a learnable parameter here (or at least is optimized at a more granular level) ? Could you at least report alpha per subset ?
> >
> > This is a valid assumption that we agree needs to be evaluated. Our concern for choosing a too granular alpha during tuning is that the method can overfit to the dev-set. To test the added benefit of more granular alphas, we additionally tested the hybrid methods in steps of 0.01 over the test-set without tuning, and found almost no improvements over the 0.1 sized steps. We aim to provide these additional results to the paper soon. We provide the alpha per subset in tables 21-24.

---

> > > ### Comment · Reviewer_EqPn · 2025-11-22
> > >
> > > Thanks for your detailed response and strong improvements to the paper. Before improving my grade, would it be possible to clarify:
> > >
> > > **Request for Clarifications**:
> > >
> > > Table 3 and Figure 3 do not seem to report the same averages. I also looked at the current ViDoRe V3 results on the MTEB leaderboard/blog and seems like there are averages over english / multilingual splits, and private / public leaderboards. This is not super clear to me... What is the "Current SOTA" that is reported ?

---

> > > > ### Author Response · Authors · 2025-11-23
> > > >
> > > > We thank the reviewer for the quick response, the appreciation of our work, and the willingness to improve the score! Regarding the questions raised:
> > > >
> > > >
> > > > > Table 3 and Figure 3 do not seem to report the same averages.
> > > >
> > > >
> > > > Figure 3 is based on the performance-latency on ViDoRe 2 (and compatible with Table 1), while Table 3 is based on ViDoRe 3. We will make this more clear in the captions and in the paper. (The main reason for this is that running the large and slow Nvidia-Nemo on ViDoRe 3 is computationally prohibitive, since the new benchmark is much larger than the previous generation)
> > > >
> > > > > I also looked at the current ViDoRe V3 results on the MTEB leaderboard/blog and seems like there are averages over english / multilingual splits, and private / public leaderboards. This is not super clear to me... What is the "Current SOTA" that is reported ?
> > > >
> > > > We thank the reviewer for this question, and we will make it clearer in the paper. The ViDoRe 3 benchmark contains 10 subsets, 8 subsets publicly released (including 3 multilingual ones) and 2 private english subsets. The performance we report is over these 8 public subsets. The MTEB leaderboard supports only a single-model evaluation protocol to the best of our knowledge, and so we can't evaluate GQR on the 2 private subsets. We do note that Colnomic and Jina-V4 are the currently leading models on the benchmark, and we improve their performance by significant margin. This blog post provides more details on the construction of ViDoRe 3 - https://huggingface.co/blog/QuentinJG/introducing-vidore-v3
> > > >
> > > >
> > > > We would be happy to answer further questions or provide additional details!

---

> ### Comment · Reviewer_EqPn · 2025-11-23
>
> Thank you for the clarifications !
>
> I believe this is a very solid rebuttal that has mostly answered my many questions. I believe the additional insights greatly improve the paper, and ViDoRe V3 0-shot results give me strong confidence that this method is broadly applicable. I would however like to see tuned score aggregation baselines (fitted on V2 in the same settings) for V3 as well which I did not see.
>
> The fact this method outperforms a tuned averaging baseline was not intuitive to me, but the authors have dissipated most of my methodological doubts (would still be key to have baselines on V3) - which makes the paper all the more interesting. I believe the paper is sound, and would be of interest to members of the community.  I will improve my soundness and contribution scores, and my overall grade to a 6 temporarily.

---

> > ### Author Response · Authors · 2025-12-03
> >
> > We again thank the reviewer for the appreciation of our work, the trust in our contributions and the interest in their value to the community. We are sorry that the discussion period was cut short and that we could not receive your final evaluation.
> >
> > > I would however like to see tuned score aggregation baselines (fitted on V2 in the same settings) for V3 as well which I did not see.
> >
> > We have updated the paper with the results of the baselines on ViDoRe 3, presented in Table 5 in the appendix, tuned on ViDoRe 2 in the exact same way as before, shown in Figure 7 in the appendix. The results show that GQR remains superior in zero shot settings over the baselines, achieving the highest average gain across model pairs while also being the most stable with the lowest standard deviation.

---

### Official Review · Reviewer_NbzC · 2025-10-29

**Soundness:** 3
**Presentation:** 2
**Contribution:** 3
**Rating:** 6
**Confidence:** 3

**Summary:**

This paper introduces Guided Query Refinement (GQR), a test-time optimization technique designed to improve visual document retrieval, by iteratively refining the query embedding of a primary retriever using guidance from a complementary retriever. Instead of aggregating outputs at the ranking or score level, GQR operates at the representation level. The method is evaluated on ViDoRe 1 and ViDoRe 2.

**Strengths:**

1. The core idea and methodology of the paper are concise, and the inclusion of algorithm pseudocode makes it easy to understand and follow.
2. The authors evaluated nine retrieval pairs on standard retrieval visual document benchmarks, demonstrating the efficiency of the proposed method in most cases.
3. The code has been made publicly available, but there appear to be some minor shortcomings.

**Weaknesses:**

1. The paper is titled Multimodal Hybrid Retrieval; however, the authors focus solely on a single visual document retrieval task. In fact, multimodal hybrid retrieval covers a much broader scope — it may also include tasks such as video-text retrieval and composed image retrieval. Therefore, the paper does not fully demonstrate that the proposed method is applicable to the broader multimodal hybrid retrieval setting.

2. The algorithm relies on tuning hyperparameters such as $\alpha$ and $T$, yet the authors do not present any analysis of the model’s sensitivity to these hyperparameters, and the paper lacks corresponding ablation studies.

3. The proposed approach is neither plug-and-play nor zero-shot. As I understand it, it requires manually collecting a sub-dataset before fine-tuning can begin. Such a high “startup cost” could hinder its scalability and practical adoption.

**Questions:**

The authors need to address the above weaknesses. In addition, I have a few further questions and suggestions.

1. If I want to use multiple complementary retrievers in parallel, how should the algorithm be modified specifically? Have the authors explored this direction? I am also curious whether the differences in query representations extracted by multiple complementary retrievers would have an impact on the final results.

2. Figure 1 does not provide sufficient information. I suggest that the authors integrate Figure 4 into the main text.

3. The visuals in Figure 2 and Figure 3 appear rather sparse and monotonous. It would be better if these figures were made more visually appealing.

---

> ### Author Response · Authors · 2025-11-22
>
> We thank the reviewer for their appreciation of our work!
> We address comments and questions below, and would be happy to engage further during the discussion phase.
>
> > The paper is titled Multimodal Hybrid Retrieval; however, the authors focus solely on a single visual document retrieval task. In fact, multimodal hybrid retrieval covers a much broader scope — it may also include tasks such as video-text retrieval and composed image retrieval. Therefore, the paper does not fully demonstrate that the proposed method is applicable to the broader multimodal hybrid retrieval setting.
>
> While we do not claim to cover all retrieval tasks that involve multimodality, we nevertheless believe that the problem of visual document retrieval is sufficiently important and broad in and of itself, and also one that is very relevant for real-world use cases. We note that this is a very active and prominent research field, as evidenced by the multitude of other studies, benchmarks and models that focus on this broad subfield within multimodal retrieval. Applying GQR to tasks like video-text retrieval or composed image retrieval is definitely an interesting avenue for future work.
>
> > The algorithm relies on tuning hyperparameters such as $\alpha$ and T, yet the authors do not present any analysis of the model’s sensitivity to these hyperparameters, and the paper lacks corresponding ablation studies.
>
> We have added a sub-section in the analysis (§4.1, Figure 4) which investigates the relation between the hyperparameter values T and $\alpha$ and the resulting GQR performance. This analysis has also informed our choice of parameter configuration for the newly-added zero-shot experiment on ViDoRe 3.
>
> > The proposed approach is neither plug-and-play nor zero-shot. As I understand it, it requires manually collecting a sub-dataset before fine-tuning can begin. Such a high “startup cost” could hinder its scalability and practical adoption.
>
> To address these concerns, we have conducted additional experiments testing the application of GQR in a zero-shot scenario as well. In the added results section “3.3 Results – Zero-Shot Performance”, **we run GQR on the newly-released ViDoRe V3 benchmark, using a single hyperparameter configuration across 6 model pairs and 8 subsets. We find that GQR is highly effective, improving performance for all pairs in a zero-shot setting, and achieves state-of-the-art performance on this benchmark without task-specific tuning.**
>
> > If I want to use multiple complementary retrievers in parallel, how should the algorithm be modified specifically? Have the authors explored this direction? I am also curious whether the differences in query representations extracted by multiple complementary retrievers would have an impact on the final results.
>
> In principle, the most straightforward way to apply the algorithm would be to calculate the consensus distribution $p_{avg}$ (line 193) not as an average of $p_1$ and $p_2$, but as an average of $p_1$ and the distributions $p_m$ of more than one complementary retrievers, and apply the rest of the GQR algorithm as-is. We have yet to explore the setting of more than one complementary retriever and agree that it would be interesting to explore the effect that such a multi-perspective setting may have on the results. At the same time, increasing the number of complementary retrievers will have a significant computational tradeoff, on latency, storage and GPU memory, which is the primary reason we have focused on a single complementary retriever in this work.
>
> > Figure 1 does not provide sufficient information. I suggest that the authors integrate Figure 4 into the main text.
>
> Following the reviewer's suggestion, we have included Figure 4 (now Figure 2) in the main text.
>
> > The visuals in Figure 2 and Figure 3 appear rather sparse and monotonous. It would be better if these figures were made more visually appealing.
>
> We thank the reviewer for the feedback. We have edited Figure 2 (now Figure 3) in an attempt to more clearly convey the information.

---

> > ### Author Response · Authors · 2025-11-27
> >
> > We would be very happy to clarify any remaining questions or discuss any points that remain unclear. We would greatly appreciate the reviewer's input and evaluation of our added results and improvements to the paper.

---

### Official Review · Reviewer_LwFe · 2025-10-30

**Soundness:** 3
**Presentation:** 2
**Contribution:** 2
**Rating:** 6
**Confidence:** 3

**Summary:**

This paper focuses on the problem of efficient visual document retrieval and proposes Guided Query Refinement (GQR), a test-time optimization framework that refines a primary retriever’s query embedding using guidance from a complementary retriever. By integrating rich cross-model interactions often missed by standard hybrid methods, GQR enables ColPali-based multimodal retrievers to achieve state-of-the-art performance. Extensive experiments demonstrate that GQR effectively advances the Pareto frontier of performance and efficiency in multimodal document retrieval.

**Strengths:**

1.	This paper innovatively adopts a vision-centric aggregation paradigm, supplemented by a text-centric approach, to address the issues of excessive modality gaps and high computational and storage costs in visual document retrieval tasks.
2.	The arguments in this paper are well-substantiated, and the experimental section is thorough and comprehensive.
3.	The description of the GQR algorithm in this paper is clear and easy to understand; even though there are many formulas, they are still easy to follow.

**Weaknesses:**

1.	The figures in this paper are somewhat rough, and Figure 4, which illustrates the core GQR architecture, should not have been placed in the appendix.
2.	The introduction does not provide sufficient background information and lacks an explanation of the necessity of applying hybrid retrieval methods.

**Questions:**

1.	The paper does not include experiments on the number of iterations T. How does this hyperparameter affect performance?
2.	In the introduction, the authors claim that the hybrid retrieval approach aims to leverage the advantages of dense text retrievers in terms of lower time and storage costs. However, the main model proposed in this paper is a multimodal, vision-centric encoder, and the dense text retriever merely provides additional supervision. So where exactly does the efficiency advantage of the hybrid retrieval paradigm manifest?

---

> ### Author Response · Authors · 2025-11-22
>
> We thank the reviewer for their appreciation of our work!
> We address comments and questions below, and would be happy to engage further during the discussion phase.
>
> > The figures in this paper are somewhat rough, and Figure 4, which illustrates the core GQR architecture, should not have been placed in the appendix.
>
> We have improved the figures in the current version and have also placed Figure 4 (now Figure 2) in the main text of the paper, as suggested.
>
> > The introduction does not provide sufficient background information and lacks an explanation of the necessity of applying hybrid retrieval methods.
>
>
> Following the reviewer's remark, we edited the section of the introduction on hybrid retrieval (line ~84), to further clarify the general motivation for applying hybrid retrieval methods. We further note that in addition to the introduction and related work sections, the appendix includes a thorough background section (Appendix A) that helps provide additional context for the reader.
>
>
> > The paper does not include experiments on the number of iterations T. How does this hyperparameter affect performance?
>
> We have added a sub-section in the analysis (§4.1, Figure 4) which investigates the relation between the hyperparameter values T and $\alpha$ and the resulting GQR performance. This analysis has also informed our choice of parameter configuration for the newly-added zero-shot experiment on ViDoRe 3. We encourage the reviewer to view section 3.3, **where we show that GQR achieves state-of-the-art performance on ViDoRe3 using a single hyperparameter configuration across 6 model-pairs, improving performance across 8 benchmark subsets.**
>
> > In the introduction, the authors claim that the hybrid retrieval approach aims to leverage the advantages of dense text retrievers in terms of lower time and storage costs. However, the main model proposed in this paper is a multimodal, vision-centric encoder, and the dense text retriever merely provides additional supervision. So where exactly does the efficiency advantage of the hybrid retrieval paradigm manifest?
>
> To clarify, our claim is about an efficiency advantage *for a given level of retrieval performance*. In the scenario of visual document retrieval, dense text retrievers on their own are indeed very efficient, but their retrieval quality is lower (as seen in the top rows in Table 1); in contrast, a leading vision-centric model like Llama-Nemo shows good retrieval quality, but has extremely high costs – even compared to other multimodal retrievers. Thus, our goal here is to reach a better trade-off, where the quality is just as good – or better – as the best vision-centric encoders out there, but the solution is significantly more efficient. In other words, we demonstrate an efficiency advantage of GQR compared to using a stronger and more expensive vision-centric model, and not compared to using an inferior (but very efficient) text model.

---

> ### Comment · Reviewer_LwFe · 2025-11-26
>
> Thank you for the author's detailed response. I am quite satisfied with it and agree to accept this paper.

---

### Official Review · Reviewer_StJj · 2025-11-01

**Soundness:** 3
**Presentation:** 2
**Contribution:** 3
**Rating:** 4
**Confidence:** 4

**Summary:**

This paper proposes Guided Query Refinement (GQR), a novel test-time hybrid retrieval method. Specifically, GQR iteratively optimizes the representation of a primary retriever using a complementary retriever as a guidance signal. The optimization is performed via gradient descent to minimize the KL divergence between the primary retriever’s distribution and a consensus distribution (initialized as the average of the primary and complementary retrievers). Experimental results show that the proposed method achieves performance comparable to SOTA models while being significantly more efficient.

**Strengths:**

1. The proposed GQR method is an effective approach to achieving an impressive performance–efficiency trade-off.
2. The experiments effectively validate the proposed method’s advantages compared to both base retrievers and hybrid baselines.
3. The finding that a weaker retriever can still provide a complementary signal to improve a stronger retriever is interesting.

**Weaknesses:**

1. The authors should clarify the “Score Aggregation – Tuned” results in Table 2. From my current understanding, the Score Aggregation method simply merges the scores from different retrievers. If this is correct, the Score Aggregation (Min-Max) – Tuned method (with just an arithmetic weighting) only drops 0.5% in performance but is much more efficient than GQR, which requires gradient optimization.
2. The hyperparameters are selected based on the task, raising concerns about generalization.
3. There is no ablation study on the impact of the step T on performance.
4. There is a lack of exploration regarding how to choose the consensus distribution and how to define the distance between the consensus and the primary retriever distribution.

**Questions:**

1. Could the authors clarify each method listed in Table 2?
2. How does the step T influence the performance of the proposed method?

---

> ### Author Response · Authors · 2025-11-22
>
> We thank the reviewer for their comments and appreciation of our work!
> Below we address each of the concerns, and we look forward to a productive discussion on how we can further improve the work.
>
>
> > The authors should clarify the “Score Aggregation – Tuned” results in Table 2. From my current understanding, the Score Aggregation method simply merges the scores from different retrievers. If this is correct, the Score Aggregation (Min-Max) – Tuned method (with just an arithmetic weighting) only drops 0.5% in performance but is much more efficient than GQR, which requires gradient optimization.
>
> The Tuned Score Aggregation baseline is indeed a weighted average of the retriever scores, after applying some normalization. We have now included the formulation in Equation 11.
> To address the reviewer's concern:
>
>
> (1) We would like to emphasize that the efficiency gain of the Score Aggregation (Min-Max) over GQR is minor compared to the overall latencies discussed in the paper. We added the online latency breakdown of GQR in section 4.2, and it shows that the latency of the gradient optimization is 50/100ms for T=25 and T=50. Models equipped with GQR achieve state-of-the-art performance while being an order-of-magnitude faster than larger models. To give a concrete example, 'Score Aggregation (Min-Max)' and 'Score Aggregation (SoftMax)'  between Colnomic and Linq, achieve 62 and 61.3 NDCG@5 respectively at 130ms. While GQR achieves 62.7 NDCG@5 at a slightly slower 181ms, it is the only method of the three that matches the performance of nvidia-nemo which averages at 2591ms per query. These performance-latency tradeoffs generally favor GQR.
>
> (2) In addition to the added performance gains compared to the baselines, in our experiments we also found GQR to perform more stably across different model pairs than the weight baseline. To demonstrate this additional benefit, we have now added to Table 2 the standard deviation for each hybrid method across model pairs. GQR is the most stable positive method in the table.
>
> > The hyperparameters are selected based on the task, raising concerns about generalization.
>
> To address these concerns, we have conducted additional experiments testing the application of GQR in a zero-shot scenario as well. In the added results section “3.3 Results – Zero-Shot Performance”, we run GQR on the newly-released ViDoRe V3 benchmark, **using a single hyperparameter configuration across 6 model pairs and 8 subsets. We find that GQR is highly effective, improving performance for all pairs in a zero-shot setting, and achieves state-of-the-art performance on this benchmark without task-specific tuning.**
>
> > How does the step T influence the performance of the proposed method?
>
> We have added a sub-section in the analysis (§4.1, Figure 4) which investigates the relation between the hyperparameter values T and $\alpha$ and the resulting GQR performance. This analysis has also informed our choice of parameter configuration for the newly-added zero-shot experiment on ViDoRe 3.
>
> > There is a lack of exploration regarding how to choose the consensus distribution and how to define the distance between the consensus and the primary retriever distribution.
>
> We added Section 2.4 to clarify the rationale behind our objective. We use KL divergence, in line with standard distillation setups, and define the consensus distribution as the average of the two retrievers because the complementary retriever is typically weaker and should not dominate the training signal. We do test alternative choices in subsection §4.4 of the paper. Specifically, we compare the use of a consensus distribution to directly using the distribution $p_2$ as the target distribution, and we also test Jensen-Shannon divergence as an alternative to KL. We additionally plan to follow up with a thorough hyperparameter investigation for these alternative objectives. If there are additional explorations the reviewer deems necessary, we would be happy to discuss this further.
>
> > Could the authors clarify each method listed in Table 2?
>
> In addition to the general overview of the ranking-based and score-based hybrid baselines given in sections §2.1 and §2.2, we have now included additional formulations in the paper so that all hybrid baselines are described more precisely. The hybrid methods from Table 2 are defined as follows: RRF - Equation 1; Average Ranking - Equation 6; Score Aggregation is described in Equation 2, where the variants use either Min-Max (Equation 7) or Softmax (Equation 8) for normalization; For the tuned variants of the hybrid baselines, we provide a general description in §2.2, and of the exact hyperparameter tuning procedure in Appendix D; Average Ranking (Tuned), RRF (Tuned) and Score Aggregation (Tuned) are now explicitly defined in Equations 9-11. Please let us know if any details remain unclear.

---

> > ### Comment · Reviewer_StJj · 2025-11-26
> >
> > Thanks for the detailed rebuttal. This rebuttal addresses most of my concerns. It's good to see the additional experiments to support the authors' statements and show generalization for GQR. I increase my rating to 6 accordingly.

---

### Author Response · Authors · 2025-12-03
**Rebuttal Summary - Part 1**

Dear Area Chair and Conference Committee Members,

We would like to emphasize that the current scores do not reflect the reviewers’ responses during the discussion period or the quality of our rebuttal. At the time of reversion, the scores were 6-6-6-6, accompanied by very positive verbal feedback from all reviewers. The scores could potentially have improved further during the discussion period, as we were awaiting revised evaluations from two reviewers after addressing their concerns. We are not aware of any weakness or concern raised by any reviewer that we did not resolve or provide substantial evidence to rebut.

We will begin by summarizing the state of the reviews at the time of reversion, and then address the concerns raised during the discussion period and explain how we have resolved them in our rebuttal and revised manuscript.


### StJj:
We satisfied the reviewer's concerns and have answered all questions.

> Thanks for the detailed rebuttal. This rebuttal addresses most of my concerns. It's good to see the additional experiments to support the authors' statements and show generalization for GQR. I increase my rating to 6 accordingly.

### EqPn:
We engaged in a detailed rebuttal with the reviewer, conducted additional experiments and satisfied the concerns:

> I believe this is a very solid rebuttal that has mostly answered my many questions. I believe the additional insights greatly improve the paper, and ViDoRe V3 0-shot results give me strong confidence that this method is broadly applicable.

> The fact this method outperforms a tuned averaging baseline was not intuitive to me, but the authors have dissipated most of my methodological doubts (would still be key to have baselines on V3) - which makes the paper all the more interesting. I believe the paper is sound, and would be of interest to members of the community. I will improve my soundness and contribution scores, and my overall grade to a 6 temporarily.

Accordingly, the reviewer increased the score to 6 with a confidence of 5. The reviewer also noted that this score was temporary, implying that it could be raised further once the baselines on the ViDoRe 3 benchmark were provided. We subsequently supplied these results and demonstrated that GQR remains superior to the baselines.


### LwFe:

We have addressed all of the reviewers' questions, and improved the figures and lacking background the reviewer pointed to.

> Thank you for the author's detailed response. I am quite satisfied with it and agree to accept this paper.

### NbzC:

We addressed all of the reviewer’s concerns and questions in our rebuttal. Since the reviewer did not respond to our rebuttal, we sent a reminder.

> We would be very happy to clarify any remaining questions or discuss any points that remain unclear. We would greatly appreciate the reviewer's input and evaluation of our added results and improvements to the paper.

At this point, the reviewer was unable to continue participating in the discussion.

---

> ### Author Response · Authors · 2025-12-03
> **Rebuttal Summary - Part 2**
>
> In this section, we summarize the main concerns and questions raised by the reviewers and explain how we resolved each of them in our rebuttal.
>
> Three primary concerns regarding GQR which were raised by the reviewers are: (1) its applicability in zero-shot settings (StJj, NbzC, EqPn), (2) the need for additional hyperparameter investigation (StJj, LwFe, NbzC, EqPn), and (3) the rationale behind the method (StJj, EqPn).
>
> (1) GQR applicability in zero-shot settings.
>
> We have conducted additional experiments testing the application of GQR in a zero-shot scenario. We run GQR on the newly-released ViDoRe V3 benchmark, **using a single hyperparameter configuration across 6 model pairs and 8 subsets. We find that GQR is highly effective, improving performance for all pairs in a zero-shot setting, and achieves state-of-the-art performance on the public subsets of the benchmark without task-specific tuning.**
> Notably, two reviewers who raised this concern verbally acknowledged that these results constituted convincing evidence, while the other didn't respond in time.
>
> (2) Additional hyperparameter investigation
>
> We have added a sub-section in the analysis (§4.1, Figure 4) which investigates the relation between the hyperparameter values T and $\alpha$ and the resulting GQR performance. We specifically show the trends for different combinations of T and alpha, and provide recommended configurations for zeroshot adaptations. In fact, this analysis has also informed our choice of parameter configuration for the newly-added zero-shot experiment on ViDoRe 3.
>
> (3) The rationale behind the GQR objective and query refinement
>
> We added Section 2.4 to clarify the rationale behind our objective. Concretely, we use KL divergence, in line with standard distillation setups, and define the consensus distribution as the average of the two retrievers because the complementary retriever is typically weaker and should not dominate the training signal. We do test alternative choices in subsection §4.4 of the paper. Specifically, we compare the use of a consensus distribution to directly using the distribution as the target distribution, and we also test Jensen-Shannon divergence as an alternative to KL.
>
>
> In Section 2.4 we also clarify the motivation for query refinement. The score updates in GQR are the result of query-document similarity with the updated query, and since the documents are fixed in this space, the scores are constrained to the original geometry. This means that the feedback of the complementary retriever is incorporated more softly and with alignment to the primary retriever. As an intuitive example, one can consider two score distributions - p_1=[0.4,0.4,0.2], p_2=[0.33,0.33,0.33]. In score-level fusion, p_1 is nudged towards p_2 in equal magnitudes for each document, so the result is always a simple weighted average (affine combination). In GQR, it is possible that the geometry of the space will nudge each document score in different magnitudes and in a non-linear way, so after GQR the scores will be p_gqr=[0.34,38,28] (e.g. the first document is easily nudged toward p_2 while the second one is not easily changed, due to the inductive bias in the primary retriever's space).
> To further provide intuition for the readers, we also added some examples of the query dynamics in Appendix G. These examples demonstrate that the effects of GQR differ substantially from those of a simple weighted average, and are often focused on a specific subset of candidate documents.
>
>
> Besides these main concerns, only minor issues were raised, such as moving a figure from the appendix into the main paper or providing additional background on hybrid retrieval in the introduction. We have addressed all of these points.

---

### Meta-Review · Area_Chair_XWKq · 2025-12-24

**Summary:**

The paper received initial scores of 4, 6, 6, and 4, which is in the borderline range for acceptance. Reviewers StJj, EqPn, and LwFe found the rebuttal satisfactory and supportive of acceptance. For the concerns raised by reviewer NbzC, I believe that if he had participated in the discussion phase, their scores would likely have been updated upward. After carefully reviewing the comments and the authors’ responses, I believe the paper meets the acceptance criteria for ICLR.

**Reviewer Concerns:**

Nearly all concerns raised by the reviewers have been adequately addressed.

**Reviewer Scores:**

For the concerns raised by reviewer NbzC, I believe that if he had participated in the discussion phase, their scores would likely have been updated upward.

---

### Decision · Program_Chairs · 2026-01-26

Accept (Poster)